# Scalable Model Merging with Progressive Layer-wise Distillation

**Jing Xu** [1 2]  **Jiazheng Li** [2 3]  **Jingzhao Zhang** [1 2]

## Abstract

Model merging offers an effective way to integrate the capabilities of multiple fine-tuned models. However, the performance degradation of the merged model remains a challenge, particularly when none or few data are available. This paper first highlights the necessity of domain-specific data for model merging by proving that data-agnostic algorithms can have arbitrarily bad worst-case performance. Building on this theoretical insight, we explore the relationship between model merging and distillation, introducing a novel few-shot merging algorithm, `ProDistill` (*Pro*gressive Layer-wise *Distill*ation). Unlike common belief that layer-wise training hurts performance, we show that layer-wise teacher-student distillation not only enhances the scalability but also improves model merging performance. We conduct extensive experiments to show that compared to existing few-shot merging methods, `ProDistill` achieves state-of-the-art performance, with up to 6.14% and 6.61% improvements in vision and NLU tasks. Furthermore, we extend the experiments to models with over 10B parameters, showcasing the exceptional scalability of `ProDistill`.

## 1. Introduction

Large-scale pre-trained models have revolutionized deep learning, achieving remarkable success across various domains such as language (Brown et al., 2020; Team et al., 2023; Touvron et al., 2023a) and vision (Dosovitskiy, 2020; Ramesh et al., 2021). Meanwhile, an increasing number of fine-tuned checkpoints are being made publicly avail-

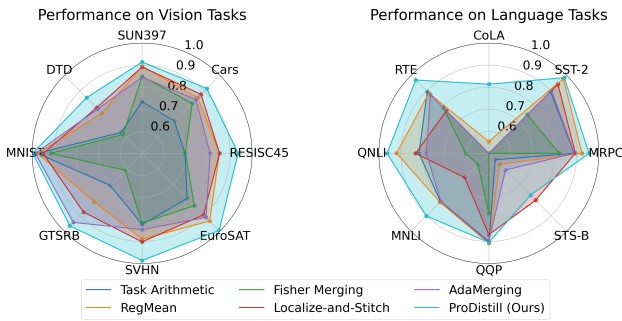

*Figure 1.* **`ProDistill` consistently outperforms other methods across nearly all considered tasks.** The performance metrics for each task are normalized and then clipped at a minimum value of 0.5 for better visualization.

able on platforms like Hugging Face. Depending on the specific downstream datasets, fine-tuned models excel in specialized abilities, such as mathematics or coding. However, complex tasks often require the integration of multiple abilities. For example, solving an advanced math problem may necessitate the assistance of computer programs to produce accurate solutions. While multi-task learning (Caruana, 1997; Misra et al., 2016; Sener & Koltun, 2018; Liu et al., 2019) can address this challenge, it requires access to fine-tuning data and incurs significant computational overhead during retraining. On the other hand, model ensembling (Dietterich et al., 2002; Kurutach et al., 2018; Ganaie et al., 2022) avoids retraining but introduces substantial storage overhead due to the need to deploy multiple models.

Model merging (Yang et al., 2024a; Goddard et al., 2024; Tang et al., 2024) offers an elegant solution to these challenges. The work of Ilharco et al. (2022) finds that the difference between fine-tuned and pre-trained weights, which they name *task vectors*, exhibits arithmetic properties, such as addition and negation, which correspond to changes in model capabilities. Therefore, model merging can be achieved by taking a weighted average of the model weights, as illustrated in Figure 2. This is connected to the linear mode connectivity (Frankle et al., 2020; Mirzadeh et al., 2020) of neural networks.

Although model merging improves storage efficiency and data protection, the performance of the merged model can degrade, especially when the number of models scales up.

---

[1]Institute for Interdisciplinary Information Sciences, Tsinghua University [2]Shanghai Qizhi Institute [3]School of Computer Science, Beijing Institute of Technology. Correspondence to: Jing Xu <xujing21@mails.tsinghua.edu.cn>, Jiazheng Li <Foreverlasting1202@outlook.com>, Jingzhao Zhang <jingzhaoz@mail.tsinghua.edu.cn>.

*Proceedings of the 42$^{st}$ International Conference on Machine Learning*, Vancouver, Canada. PMLR 267, 2025. Copyright 2025 by the author(s).

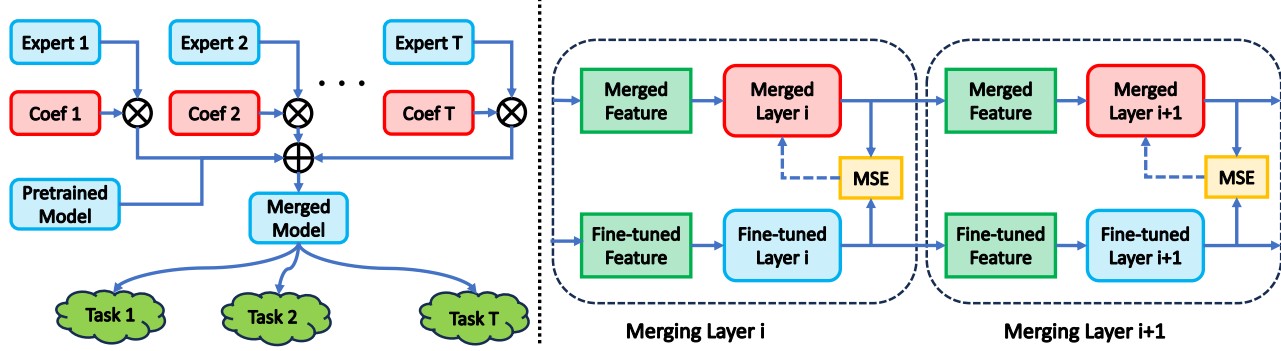

*Figure 2.* **Left: Overview of model merging.** Each expert corresponds to a task vector $\boldsymbol{\theta}_i - \boldsymbol{\theta}_0$, which is scaled by its corresponding merging coefficient $\boldsymbol{\lambda}_i$ and summed to get the merged model. **Right: Illustration of `ProDistill`.** The merged model layer and each fine-tuned model layer take as input the merged feature and the fine-tuned feature, respectively. The MSE loss between these outputs is used to update the merged model layer. The output features serve as inputs for merging the subsequent layer.

Recent studies (Matena & Raffel, 2022; Jin et al., 2022; Yang et al., 2023) propose various methods to handle this issue, many of which require a *few-shot validation dataset* that contains domain-specific information of downstream tasks. This seems to contradict the data-free nature of task arithmetic. In light of this, we raise the question:

> *Is domain-specific data necessary for model merging?*

We provide an affirmative theoretical answer to this question. We prove, for the first time, that the *worst-case* performance of *any data-agnostic* model merging algorithm can be arbitrarily bad, even for a simple linear model. Therefore, although data-agnostic algorithm achieves great empirical success, it is theoretically reasonable to assume access to at least a few-shot dataset.

Building on these theoretical insights, we next address the empirical challenge of few-shot model merging by exploring the following question:

> *How to fully leverage the data and the fine-tuned models to improve merging performance?*

To this end, we frame model merging as a teacher-student distillation problem, where the goal is to transfer the knowledge from multiple fine-tuned models (teacher) into the merged model (student). However, directly applying existing distillation algorithms often results in large training memory overhead, particularly for large language models with billions of parameters.

To address this challenge, we propose a novel model merging algorithm, `ProDistill` (*Pro*gressive Layer-wise *Distill*ation). `ProDistill` implements distillation through an activation matching strategy, where the merging coefficients are trained to minimize the activation distance between teacher and student models. The training objective

in `ProDistill` is decomposed layer by layer (Bengio et al., 2006; Kulkarni & Karande, 2017; Hettinger et al., 2017; Karkar et al., 2024; Sakamoto & Sato, 2024), enabling the algorithm to avoid traditional end-to-end training and instead progressively train each layer of the model. See Figure 2 for an illustration.

We conduct extensive experiments to evaluate the performance of `ProDistill` across various tasks, architectures, and scales. [1] Compared to both training-based and training-free baselines, `ProDistill` achieves a notable 6.14% increase in absolute performance for vision tasks and 6.61% increase for natural language understanding tasks.

Furthermore, `ProDistill` demonstrates improved data and computation efficiency, and incurs significantly lower memory costs. This makes `ProDistill` scalable to large model sizes. We apply `ProDistill` to merge large language models with over 10B parameters. To the best of our knowledge, this is the first time a *training-based* merging algorithm has been scaled to such a large model size.

We summarize our contribution as follows:

- We provide the first theoretical analysis on the necessity of domain-specific data for model merging, proving its critical role in ensuring effective merging performance.

- We propose `ProDistill`, a novel model merging algorithm that leverages teacher-student distillation to progressively merge model layers.

- We conduct comprehensive empirical analyses to demonstrate the state-of-the-art performance of `ProDistill` on a wide variety of tasks. Our experiments highlight the data, computation, and memory efficiency of the proposed method.

[1]Code is available at https://github.com/JingXuTHU/Scalable_Model_Merging_with_Progressive_Layerwise_Distillation.

## 2. Preliminaries

We consider model merging in a pretrain-to-finetune setup. Let $\boldsymbol{\theta}_0$ denote the weights of a pre-trained model. Consider a set of $T$ tasks, each with a model $\boldsymbol{\theta}_i$ fine-tuned from $\boldsymbol{\theta}_0$. Model merging aims to combine the knowledge learned by task-specific models $\boldsymbol{\theta}_i$ into a unified model $\hat{\boldsymbol{\theta}}$, which preserves the generalization ability of the pre-trained model and incorporates the specialized knowledge from each task.

**Task vectors.** A key insight in this setup is the task vectors. The *task vector* for the $i$-th task is defined as $\boldsymbol{\tau}_i = \boldsymbol{\theta}_i - \boldsymbol{\theta}_0$. An effective model merging method (Ilharco et al., 2022; Zhang et al., 2023) is to compute a weighted average of task vectors and add it back to the pre-trained model:

$$\hat{\boldsymbol{\theta}} = \boldsymbol{\theta}_0 + \sum_{i=1}^{T} \boldsymbol{\lambda}_i \circ \phi(\boldsymbol{\tau}_i),$$

where $\boldsymbol{\lambda}_i$ denotes the merging coefficients, and $\phi(\cdot)$ is an optional transformation function applied to the task vectors.

The merging coefficients $\boldsymbol{\lambda}_i$ can operate at different granularities. Common approaches include task-wise granularity (Ilharco et al., 2022), which assigns a single merging coefficient to each task, and layer-wise granularity (Yang et al., 2023), which assigns a coefficient to each layer of the models. In this paper, we take one step forward and consider **element-wise granularity**.[2] Specifically, the merging coefficient $\boldsymbol{\lambda}_i$ has the same dimensionality as $\boldsymbol{\theta}_i$, and an element-wise multiplication $\boldsymbol{\lambda}_i \circ \boldsymbol{\tau}_i$ is performed for each task. In this paper, we do not apply additional transformations $\phi(\cdot)$ to the task vectors, as our method is parallel to the transformation-based methods.

**Notations.** We use $\mathcal{D}_i$ to denote a few-shot unlabeled validation dataset for each task that is possibly available. Define $\varphi(\boldsymbol{\theta}, \cdot)$ as the feature mapping of the model parameterized by $\boldsymbol{\theta}$, which gives the vectorized embedding of all intermediate layers. Let $L$ denote layer number. For model weights $\boldsymbol{\theta}$ and layer index $l$, we use $\boldsymbol{\theta}^{(l)}$ to denote the parameter of the $l$-th layer and $\varphi^{(l)}(\boldsymbol{\theta}^{(l)}, \cdot)$ to denote the feature mapping function defined by this layer.

## 3. Theoretical Limitations on Data-Agnostic Model Merging

Model merging algorithms can be broadly classified into two categories based on data availability: *data-agnostic* algorithms, which only use the weights of pre-trained and fine-tuned models (*e.g.*, Ilharco et al. (2022); Yadav et al. (2024); Yu et al. (2024b)), and *data-dependent* algorithms, which require access to a validation set (*e.g.*, Matena & Raffel (2022); Jin et al. (2022)). While data-agnostic algo-

rithms have shown significant empirical success, we prove that their *worst-case* performance can be arbitrarily poor, even for simple linear models.

### 3.1. Hardness Results for Fixed Models

Consider the following simplified setup for model merging. Suppose we have two tasks with datasets $\mathcal{D}_1, \mathcal{D}_2$ and loss function $\ell(\cdot, \cdot)$. Let $f_1, f_2$ denote two models to merge, and let $\mathcal{M}$ denote a *data-agnostic* model merging algorithm, which we assume to be deterministic for simplicity.

The following hardness result states that for any such algorithm $\mathcal{M}$, one can always construct adversarial datasets such that the merging performance is arbitrarily bad.

**Theorem 3.1.** *There exist a task and loss function $\ell$, such that for any data-agnostic model merging algorithm $\mathcal{M}$, any pair of models $f_1 \neq f_2$, and any $\varepsilon, C > 0$, there exists two datasets $\mathcal{D}_1, \mathcal{D}_2$, such that $f_1, f_2$ have a near-zero loss on $\mathcal{D}_1$ and $\mathcal{D}_2$, respectively:*

$$\ell(D_1, f_1) \leq \varepsilon, \quad \ell(D_2, f_2) \leq \varepsilon,$$

*but the merged model $\hat{f} = \mathcal{M}(f_1, f_2)$ has a constant loss on $\mathcal{D}_1 \cup \mathcal{D}_2$:*

$$\ell(\mathcal{D}_1 \cup \mathcal{D}_2, \hat{f}) \geq C.$$

*On the other hand, there exists a ground truth model $f^*$ that has near-zero loss on $\mathcal{D}_1$ and $\mathcal{D}_2$:*

$$\ell(\mathcal{D}_1 \cup \mathcal{D}_2, f^*) \leq \varepsilon.$$

Theorem 3.1 is proved by adversarially constructing linear regression instances based on the merged model. The complete proofs are deferred to Appendix A.

### 3.2. Hardness Results for Learned Models

Theorem 3.1 assumes fixed models $f_1$ and $f_2$, which can deviate from real-world scenarios where models are trained on datasets. To address this, we extend the analysis to cases where models are learned using an algorithm $\mathcal{L}$, with $f_1 = \mathcal{L}(\mathcal{D}_1), f_2 = \mathcal{L}(\mathcal{D}_2)$. We prove the following result.

**Theorem 3.2.** *There exist a task, a loss function $\ell$ and a learning algorithm $\mathcal{L}$, such that for any data-agnostic model merging algorithm $\mathcal{M}$ and any $\varepsilon, C > 0$, there exist two adversarial datasets $\mathcal{D}_1, \mathcal{D}_2$, such that $f_1 = \mathcal{L}(\mathcal{D}_1), f_2 = \mathcal{L}(\mathcal{D}_2)$ have a near-zero loss on $\mathcal{D}_1$ and $\mathcal{D}_2$ respectively:*

$$\ell(D_1, f_1) \leq \varepsilon, \quad \ell(D_2, f_2) \leq \varepsilon,$$

*but the merged model $\hat{f} = \mathcal{M}(f_1, f_2)$ has a constant loss on $\mathcal{D}_1 \cup \mathcal{D}_2$:*

$$\ell(\mathcal{D}_1 \cup \mathcal{D}_2, \hat{f}) \geq C.$$

---

[2]The *layer-wise* in the title and algorithm name does not refer to the granularity of $\boldsymbol{\lambda}_i$, but instead refers to the training procedure.

**Algorithm 1:** `ProDistill` (*Pro*gressive Layer-wise *Distill*ation)

**Input:** Pre-trained model weights $\boldsymbol{\theta}_0$, Fine-tuned model weights $\boldsymbol{\theta}_i$, unlabeled validation sets $\{\mathcal{D}_i\}_{i=1}^{T}$.

**Output:** Merging coefficients $\hat{\boldsymbol{\lambda}}_i^{(l)}$, $1 \le i \le T$, $1 \le l \le L$.

**Initialize** $\mathcal{D}_i^{(0)} = \{(\boldsymbol{x}, \boldsymbol{x}) : \boldsymbol{x} \in \mathcal{D}_i\}$, and $\boldsymbol{\tau}_i = \boldsymbol{\theta}_i - \boldsymbol{\theta}_0$ for $i = 1, \ldots, T$.

**for** $l = 1$ **to** $L$ **do**

  **Solve the objective function using gradient descent:**

$$\{\hat{\boldsymbol{\lambda}}_i^{(l)}\}_{i=1}^{T} = \underset{\{\boldsymbol{\lambda}_i^{(l)}\}_{i=1}^{T}}{\arg\min} \sum_{i=1}^{T} \frac{1}{2T|\mathcal{D}_i|} \sum_{(\boldsymbol{z}_1, \boldsymbol{z}_2) \in \mathcal{D}_i^{(l-1)}}$$

$$\left\| \varphi^{(l)}\left(\boldsymbol{\theta}_0^{(l)} + \sum_{j=1}^{T} \boldsymbol{\lambda}_j^{(l)} \circ \boldsymbol{\tau}_j^{(l)}, \boldsymbol{z}_1\right) - \varphi^{(l)}\left(\boldsymbol{\theta}_i^{(l)}, \boldsymbol{z}_2\right) \right\|^2$$

  **for** $i = 1$ **to** $T$ **do**

    **Update** $\mathcal{D}_i^{(l)}$ **using:**

$$\mathcal{D}_i^{(l)} = \Big\{ \Big(\varphi^{(l)}\big(\boldsymbol{\theta}_0^{(l)} + \sum_{j=1}^{T} \hat{\boldsymbol{\lambda}}_j^{(l)} \circ \boldsymbol{\tau}_j^{(l)}, \boldsymbol{z}_1\big),$$

$$\varphi^{(l)}\big(\boldsymbol{\theta}_i^{(l)}, \boldsymbol{z}_2\big)\Big) : (\boldsymbol{z}_1, \boldsymbol{z}_2) \in \mathcal{D}_i^{(l-1)} \Big\}$$

  **end**

**end**

**return** $\hat{\boldsymbol{\lambda}}_i^{(l)}$ for $1 \le i \le T$, $1 \le l \le L$.

---

*On the other hand, the model learned on the merged dataset $f^* = \mathcal{L}(\mathcal{D}_1 \cup \mathcal{D}_2)$ that has near-zero loss on $\mathcal{D}_1$ and $\mathcal{D}_2$:*

$$\ell(\mathcal{D}_1 \cup \mathcal{D}_2, f^*) \le \varepsilon.$$

The hard instance constructed in the proof involves solving a linear-separable classification problem with max-margin classifiers. A broad range of algorithms fall into this category, including traditional support vector machine (SVM) (Cortes, 1995) and gradient descent algorithms that have max-margin implicit bias (Nacson et al., 2019; Lyu & Li, 2019). The key insight is that the learning process often discards certain information, such as distant data points outside the margin. This enables adversarial manipulation of the datasets to degrade merging performance.

*Remark* 3.3. *Theorem 3.1 and 3.2 are both worst-case analyses. They highlight the fundamental limitations of data-agnostic model merging algorithms, but do not contradict their empirical success. Instead, these results underscore the importance of data availability in achieving robust and consistent merging performance.*

## 4. A Practical Algorithm for Few-Shot Model Merging

In the previous section, we prove that data availability is crucial for effective model merging. Next, we propose a practical merging algorithm designed for such data-available settings. We start with a naïve distillation algorithm that directly minimizes the embedding distance, and build upon it to develop the main algorithm of this paper.

### 4.1. Model Merging as Knowledge Distillation

Model merging can be viewed through the lens of knowledge distillation, a perspective that remains underexplored within the community. In this context, the teacher models correspond to fine-tuned models, and the goal is to create a student model that integrates their knowledge and performs well on the downstream tasks.

A common strategy in knowledge distillation is to align the internal features of the teacher and student models (Young et al., 2022; Jin et al., 2024). This coincides with recent findings in the model merging community, which show that the performance of the merged model $\hat{\boldsymbol{\theta}}$ is positively correlated with the similarity between its embeddings $\varphi(\hat{\boldsymbol{\theta}}, \boldsymbol{x})$ and those of the fine-tuned models $\varphi(\boldsymbol{\theta}_i, \boldsymbol{x})$. (Zhou et al., 2023; Yang et al., 2024b). Building on this insight, we propose to align the merged model and fine-tuned models in the feature space by solving the following problem:

$$\min_{\boldsymbol{\lambda}_1, \cdots \boldsymbol{\lambda}_T} \sum_{i=1}^{T} \frac{1}{2T|\mathcal{D}_i|} \sum_{\boldsymbol{x} \in \mathcal{D}_i}$$

$$\left\| \varphi\left(\boldsymbol{\theta}_0 + \sum_{j=1}^{T} \boldsymbol{\lambda}_i \circ \boldsymbol{\tau}_i, \boldsymbol{x}\right) - \varphi\left(\boldsymbol{\theta}_i, \boldsymbol{x}\right) \right\|^2. \quad (1)$$

The objective minimizes the $\ell_2$ distance between the internal embeddings of the merged model $\boldsymbol{\theta}_0 + \sum_{j=1}^{T} \boldsymbol{\lambda}_i \circ \boldsymbol{\tau}_i$ and those of the fine-tuned model $\boldsymbol{\theta}_i$, over the unlabeled validation set $\mathcal{D}_i$ for each task. Intuitively, this encourages the merged model to behave similarly to each fine-tuned model, in its specific input domain. The objective can be solved using standard optimization algorithms such as Adam (Kingma, 2014) or AdamW (Loshchilov, 2017) on the merging coefficients $\boldsymbol{\lambda}_i$.

Throughout this paper, we refer to directly minimizing Objective 1 as `DistillMerge`. Despite its simplicity, it offers several insights:

1. **Adaptive Merging Coefficient.** Gradient descent on $\boldsymbol{\lambda}_i$ can be interpreted as selecting the appropriate merging coefficients, whose empirical importance has been repeatedly highlighted in recent works (Jin et al., 2022; Yang et al., 2023; Gauthier-Caron et al., 2024). Notably, the element-

wise granularity of $\boldsymbol{\lambda}_i$ gives the merged model greater expressive power to fit the objective.

2. **Fine-grained Model Merging via Distillation.** Minimizing 1 can also be viewed as a way to distill the knowledge from the fine-tuned teacher models into a merged student model. Unlike standard distillation algorithms (Hinton, 2015), Objective 1 leverages task vectors $\boldsymbol{\tau}_i$ as a prior on the trainable parameters. To justify this design choice, we further show in Appendix C.3 that *in the few-shot setup*:

- Feature-based distillation loss provides a stronger supervision compared with logit-based distillation;

- Optimizing the scaling coefficients yields better results, compared with directly optimizing the model weights.

### 4.2. Efficient Implementation by Progressive Layer-wise Distillation

While Equation 1 is a reasonable objective for training the merging coefficients, directly optimizing this objective incurs significant memory overhead. This is because both the task vectors and the trainable merging coefficients, which have the same dimensionality as the model parameters, have to be stored in memory. The memory cost scales linearly with the number of tasks. Besides, the optimization process requires to store the activations, gradients and optimizer states, further exacerbating memory overhead. This challenge becomes particularly critical when merging large language models, which often contain billions of parameters.

To mitigate this issue, we propose the following surrogate to Objective 1. Instead of optimizing the global objective across all layers simultaneously, we adopt a progressive, layer-by-layer merging strategy. For each layer $l$ ($1 \leq l \leq L$), we minimize the feature distance between layer embeddings using the following objective:

$$\min_{\boldsymbol{\lambda}_1^{(l)}, \cdots, \boldsymbol{\lambda}_T^{(l)}} \sum_{i=1}^{T} \frac{1}{2T|\mathcal{D}_i|} \sum_{(\boldsymbol{z}_1, \boldsymbol{z}_2) \in \mathcal{D}_i^{(l-1)}}$$
$$\left\| \varphi^{(l)}\left(\boldsymbol{\theta}_0^{(l)} + \sum_{j=1}^{T} \boldsymbol{\lambda}_i^{(l)} \circ \boldsymbol{\tau}_i^{(l)}, \boldsymbol{z}_1\right) - \varphi^{(l)}\left(\boldsymbol{\theta}_i^{(l)}, \boldsymbol{z}_2\right) \right\|^2. \quad (2)$$

Compared to Objective 1, this layer-wise formulation focuses only on minimizing the embedding distances in each layer, instead of all intermediate embeddings simultaneously. Moreover, this objective introduces **dual inputs** by feeding different intermediate embeddings to the fine-tuned and merged models. Specifically, $\mathcal{D}_i^{(l)}$ maintains pairs of embeddings $(\boldsymbol{z}_1, \boldsymbol{z}_2)$ after the $l$-th layer, where $\boldsymbol{z}_1$ is the embedding of the merged model, while $\boldsymbol{z}_2$ is the embedding of the fine-tuned model. These internal activations are cached and updated using the trained coefficients for each layer by

the following rule:

$$\mathcal{D}_i^{(0)} = \{(\boldsymbol{x}, \boldsymbol{x}) : \boldsymbol{x} \in \mathcal{D}_i\},$$
$$\mathcal{D}_i^{(l)} = \left\{ \left( \varphi^{(l)}\left(\boldsymbol{\theta}_0^{(l)} + \sum_{j=1}^{T} \hat{\boldsymbol{\lambda}}_i^{(l)} \circ \boldsymbol{\tau}_i^{(l)}\right), \boldsymbol{z}_1 \right), \right.$$
$$\left. \varphi^{(l)}\left(\boldsymbol{\theta}_i^{(l)}, \boldsymbol{z}_2\right) \right) : (\boldsymbol{z}_1, \boldsymbol{z}_2) \in \mathcal{D}_i^{(l-1)} \right\}, \quad l \geq 1.$$

This design better approximates the global distillation objective 1, **distinguishing it from previous merging algorithms based on feature alignment** (Jin et al., 2022; Yang et al., 2024b; Dai et al., 2025), which align the output of merged model and fine-tuned model *under the same input*. As demonstrated in Appendix C.4, the incorporation of dual inputs is critical for achieving high performance in layer-wise training.

We refer to this algorithm as `ProDistill`, short for *Pro*gressive Layer-wise *Distill*ation. The pseudocode for `ProDistill` is given in Algorithm 1. Compared to `DistillMerge`, `ProDistill` offers substantial efficiency gains. When merging a specific layer, `ProDistill` only requires memory for the task vector and merging coefficients of the current layer, rather than the entire model. Furthermore, the forward and backward passes are also restricted within individual layers. Interestingly, unlike the common belief that layer-wise training leads to performance degradation, we show in Appendix C.2 that `ProDistill` outperforms its end-to-end counterpart `DistillMerge`.

## 5. Experiments

In this section, we present comprehensive experiment results to evaluate the effectiveness of `ProDistill` across various settings.

### 5.1. Setup

We consider three main experimental setups: (1) Merging Vision Transformers (Dosovitskiy, 2020) on image classification tasks; (2) Merging BERT (Devlin, 2018) and RoBERTa (Liu, 2019) models on natural language understanding (NLU) tasks; (3) Merging LLAMA2 (Touvron et al., 2023b) model on natural language generation (NLG) tasks.

**Tasks and Models:** For image classification tasks, we follow the setting in Ilharco et al. (2022) and use Vision Transformer (ViT) models pre-trained on the ImageNet dataset and subsequently fine-tuned on 8 downstream datasets. For NLU and NLG tasks, we merge the BERT-base and RoBERTa-base models fine-tuned on 8 NLU tasks from the GLUE (Wang, 2018) benchmark, and perform pairwise

*Table 1.* **Performance of merging ViT-B-32 models across eight downstream vision tasks.** `ProDistill` consistently outperforms the baselines under different data availability. The results for Localize-and-Stich are directly taken from He et al. (2024).

| Method | SUN397 | Cars | RESISC45 | EuroSAT | SVHN | GTSRB | MNIST | DTD | Avg |
|---|---|---|---|---|---|---|---|---|---|
| Individual | 75.34 | 77.73 | 95.98 | 99.89 | 97.46 | 98.73 | 99.69 | 79.36 | 90.52 |
| Task Arithmetic | 55.32 | 54.98 | 66.68 | 78.89 | 80.21 | 69.68 | 97.34 | 50.37 | 69.18 |
| RegMean | 67.47 | 66.63 | 81.75 | 93.33 | 86.68 | 79.92 | 97.30 | 60.16 | 79.15 |
| Fisher merging | 63.95 | 63.84 | 66.86 | 83.48 | 79.54 | 60.11 | 91.27 | 49.36 | 69.80 |
| Localize-and-Stich | 67.20 | 68.30 | 81.80 | 89.40 | 87.90 | 86.60 | 94.80 | 62.90 | 79.90 |
| AdaMerging | 63.69 | 65.74 | 77.65 | 91.00 | 82.48 | 93.12 | 98.27 | 62.29 | 79.28 |
| `ProDistill` (Ours) | **68.90** | **71.21** | **89.89** | **99.37** | **96.13** | **95.29** | **99.46** | **68.03** | **86.04** |

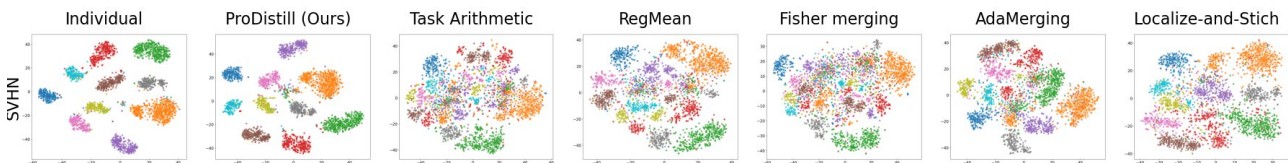

*Figure 3.* **The t-SNE visualization of ViT-B-32 model trained by different merging algorithms, on the SVHN dataset.** The features given by `ProDistill` are the most separated, resembling those of fine-tuned models.

merging of WizardLM-13B, WizardMath-13B and llama-2-13b-code-alpaca models, following the setting in (Yu et al., 2024b). Detailed information on the models and datasets can be found in Appendix B.1.

**Baselines:** For vision and NLU tasks, we compare `ProDistill` with a wide range of baselines, including Task Arithmetic (Ilharco et al., 2022), Fisher merging (Matena & Raffel, 2022), RegMean (Jin et al., 2022), AdaMerging (Yang et al., 2023) and Localize-and-Stich (He et al., 2024). All methods, except Task Arithmetic, require a few-shot unlabeled validation dataset, which is randomly sampled from the training set, with validation shot set to 64 per task. For NLG tasks, we compare `ProDistill` with Task Arithmetic (Ilharco et al., 2022), TIES-Merging (Yadav et al., 2024) and WIDEN (Yu et al., 2024a), due to scale constraints. A detailed discussion of the baselines and their implementations is provided in Appendix B.2 and B.3.

### 5.2. Results on Merging ViT models

Table 1 presents the performance of merging ViT-B-32 models across eight downstream vision tasks. The results for ViT-B-16 and ViT-L-14 are provided in Appendix D.1.

Our method consistently outperforms all baselines, yielding significant improvements in average performance. Specifically, `ProDistill` achieves an average performance of 86.04%, surpassing the baselines by 6.14%. Notably, it is only 4% below the average performance of the individual fine-tuned models.

We also visualize the final-layer activations of the merged model using t-SNE (Van der Maaten & Hinton, 2008). The results are given in Figure 3 and Appendix D.4. The visualization shows that the features given by `ProDistill` are more separated compared to the baselines, closely resembling those of the fine-tuned models.

### 5.3. Results on Merging Encoder-based Language Models

Table 2 summarizes the results of merging RoBERTa models fine-tuned on the NLU tasks. The results of BERT models are deferred to Appendix D.1. Similar to the vision tasks, `ProDistill` achieves significant performance improvements of 6.61% on the NLU tasks, outperforming all baselines across nearly all tasks.

Unlike vision tasks, the NLU tasks in the GLUE benchmark have small class numbers. For example, SUN387 dataset consists of 397 classes, while CoLA only has 2 classes. This class size disparity limits the performance of methods that operate directly on the model output logits, such as AdaMerging and Fisher merging. Our method, along with RegMean, performs particularly well, emphasizing the importance of leveraging internal feature embeddings for effective model merging.

### 5.4. Results on Merging Large Language Models

We present the results of merging the WizardMath-13B and Llama-2-13B-Code-Alpaca models in Table 3, with ad-

*Table 2.* **Performance of merging RoBERTa models on the NLU tasks.** `ProDistill` achieves superior performance across almost all tasks.

| Method | CoLA | SST-2 | MRPC | STS-B | QQP | MNLI | QNLI | RTE | Avg |
|---|---|---|---|---|---|---|---|---|---|
| Individual | 0.5458 | 0.9450 | 0.8858 | 0.9030 | 0.8999 | 0.8710 | 0.9244 | 0.7292 | 0.8380 |
| Task Arithmetic | 0.0804 | 0.8475 | 0.7865 | 0.4890 | 0.8133 | 0.7063 | 0.7558 | 0.6534 | 0.6415 |
| RegMean | 0.3022 | 0.9255 | 0.8183 | 0.5152 | **0.8176** | 0.7089 | 0.8503 | 0.6462 | 0.6980 |
| Fisher merging | 0.1633 | 0.7064 | 0.7264 | 0.1274 | 0.6962 | 0.4968 | 0.5599 | 0.5776 | 0.5068 |
| Localize-and-Stich | 0.0464 | 0.8922 | 0.7916 | **0.7232** | 0.7821 | 0.5709 | 0.7703 | 0.5632 | 0.6425 |
| AdaMerging | 0.000 | 0.8532 | 0.7875 | 0.5483 | 0.8086 | 0.7039 | 0.7247 | 0.6390 | 0.6332 |
| `ProDistill` (Ours) | **0.4442** | **0.9312** | **0.8464** | 0.6942 | 0.8134 | **0.7857** | **0.8900** | **0.7076** | **0.7641** |

ditional results provided in Appendix D.1 and generation examples provided in Appendix D.5. These findings demonstrate that our method effectively scales up to models with over 10B parameters, and achieves superior performance compared to baselines.

# 6. Further Analyses

In this section, we provide additional analyses on the efficiency of `ProDistill`, in terms of data usage, computational cost, and memory requirements.

Additionally, we conduct a broad range of comparisons and ablation studies to further evaluate `ProDistill`. The results can be found in Appendix C, including:

- Analyses of merging coefficient granularity (Appendix C.1)

- Comparison with the end-to-end merging algorithm `DistillMerge` (Appendix C.2)

- Comparison with standard supervised training and knowledge distillation (Appendix C.3)

- Ablation studies on the design of dual inputs (Appendix C.4)

## 6.1. Data Efficiency

We first evaluate the data efficiency of `ProDistill` by varying the number of validation samples, ranging from 1 to 256. The results are presented in Figure 4.

For vision tasks, `ProDistill` achieves a performance improvement of over 7% even with just 1-shot validation data per task. As the number of validation shots increases, the performance continues to improve, consistently surpassing that of AdaMerging. With 256 validation shot, the accuracy reaches 88.05%, which is only 2% lower than that of individual checkpoints.

## 6.2. Computation Efficiency

We evaluate the computational efficiency of `ProDistill` by varying the training epochs from 1 to 100. We set the validation shot to 64, and choose learning rate from {0.1, 0.01}.

The results, provided in Figure 4, highlight the rapid convergence of `ProDistill`. With just one epoch, the average accuracy of the merged model has a significant improvement, rising from 69.8 to 80.6. After approximately 10 epochs, the accuracy is nearly identical to the final results. Thus, despite being a training-based algorithm, `ProDistill` demonstrates exceptional computational efficiency with fast convergence.

The computation efficiency can be partially attributed to `ProDistill`'s ability to leverage large learning rates effectively, which we hypothesize is due to its layer-wise training scheme. In contrast, algorithms such as AdaMerging exhibit unstable convergence at high learning rates, as shown in the figure.

## 6.3. Memory Efficiency

Next, we evaluate the memory efficiency of `ProDistill` by profiling the maximum GPU memory consumption during training (excluding pre-processing and evaluation). The batch size is set to 32 for vision tasks and 16 for NLU tasks. The validation shot is set to 64. We compare `ProDistill` with its unoptimized direct training version `DistillMerge` and AdaMerging.

The results, shown in Figure 4, illustrate that `ProDistill` has an almost negligible GPU memory footprint compared to the baselines. This difference is particularly evident for models with a large number of layers, such as ViT-L-14, since the memory cost of our method remains independent of the model's depth. Therefore, `ProDistill` offers substantial advantages in memory efficiency and is scalable in resource-constrained environments.

*Table 3.* **Performance of merging LLM models on Code and Math tasks.** Our method demonstrates an improved performance and a strong scalability. The results of TIES-Merging and WIDEN are directly taken from Yu et al. (2024a).

| Method | GSM8K | MATH | HumanEval | MBPP | Avg | Norm Avg |
|---|---|---|---|---|---|---|
| WizardMath-13B | 0.6361 | 0.1456 | 0.0671 | 0.0800 | 0.2322 | 0.6430 |
| Llama-2-13b-code-alpaca | 0.000 | 0.000 | 0.2378 | 0.2760 | 0.1285 | 0.5000 |
| Task Arithmetic | **0.6467** | **0.1462** | 0.0854 | 0.0840 | 0.2406 | 0.6711 |
| TIES-Merging | 0.6323 | 0.1356 | 0.0976 | 0.2240 | 0.2723 | 0.7868 |
| WIDEN | 0.6422 | 0.1358 | 0.0976 | 0.0980 | 0.2434 | 0.6769 |
| `ProDistill` (Ours) | 0.6279 | 0.1424 | **0.1280** | **0.2239** | **0.2806** | **0.8288** |

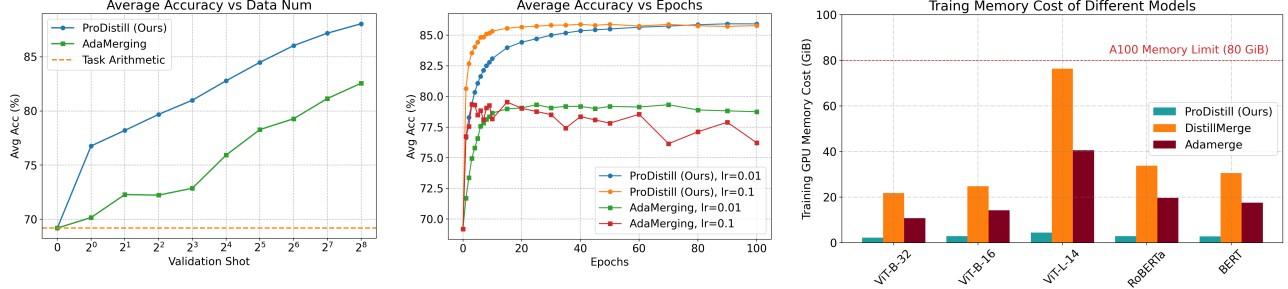

*Figure 4.* **Analysis of Data, Computation and Memory Efficiency. Left:** The average accuracy of `ProDistill` and AdaMerging across 8 vision tasks, with different data availability. Our method demonstrates superior data efficiency. **Middle:** The average accuracy of `ProDistill` with different training epochs. Our algorithm achieves a fast convergence. **Right:** The training GPU memory cost of `ProDistill`, its unoptimized counterpart `DistillMerge` and AdaMerging. Our method has a significantly smaller memory footprint.

## 7. Related Works

**Model Merging via Weight Averaging.** Weight averaging is an effective and widely adopted approach in model merging (Izmailov et al., 2018; Wortsman et al., 2022; Ilharco et al., 2022). Researchers have developed various methods to improve the averaging approach. One line of work focuses on minimizing conflicts and promoting disentanglement between task vectors, through sparsification (Tang et al., 2023a; Yadav et al., 2024; Yu et al., 2024b; He et al., 2024; Wang et al., 2024b; Bowen et al., 2024; Deng et al., 2024; Zhu et al., 2024; Davari & Belilovsky, 2025) or decomposition (Tam et al., 2023; Xiong et al., 2024; Stoica et al., 2024; Wei et al., 2025; Gargiulo et al., 2024; Marczak et al., 2025; Yang et al., 2025). Another line of work (Ortiz-Jimenez et al., 2024; Tang et al., 2023b) employs linearized training to explicitly enforce linearity. Some studies explore methods for selecting optimal merging coefficients, using training-based (Yang et al., 2023; Gauthier-Caron et al., 2024; Nishimoto et al., 2024) or training-free (Matena & Raffel, 2022; Jin et al., 2022; Zhou et al., 2024; Wang et al., 2024a; Liu et al., 2024; Tang et al., 2025) approaches. Broadly speaking, our paper aligns with the former category. Other works (Qi et al., 2024; Lu et al., 2024; Zheng & Wang, 2024; Oh et al., 2024; Zhang et al., 2024b; Osial et al.,

2024; Huang et al., 2024) propose dynamic model merging through task-specific routing and mixture-of-experts frameworks. **Our method differs from them by preserving the original model architecture**.

In addition to merging fine-tuned models, there is a broader field of research exploring more general setups, such as merging independently trained models (Singh & Jaggi, 2020; Ainsworth et al., 2022; Navon et al., 2023; Horoi et al., 2024; Stoica et al., 2023; Xu et al., 2024) and merging models with different architectures (Avrahami et al., 2022; Wan et al., 2024a;b).

**Distillation and Activation Alignment.** Knowledge distillation (Hinton, 2015; Romero et al., 2014; Yim et al., 2017) is a well-established topic in machine learning where a student model is trained to mimic the behavior of teacher models. Some works leverage teacher-student activation matching to merge models and do multi-task learning (Li & Bilen, 2020; Ghiasi et al., 2021; Yang et al., 2022; Jin et al., 2022; Kong et al., 2024; Zhang et al., 2024a; Nasery et al., 2024), which share similarities with our paper. For example, the **Surgery** algorithm proposed in (Yang et al., 2024b;c) introduces lightweight, task-specific modules to facilitate post-merging activation matching. However, these methods differ from ours as dynamic model merging approaches. **A**

**very recent work** Dai et al. (2025) matches the activations in each layer by solving linear equations of merging coefficient. Their approach differs from ours in several key aspects. First, their merging coefficient granularity is layerwise, whereas we consider element-wise coefficients. This distinction is crucial, as their method cannot be directly extended to element-wise merging without making the linear equations under-determined. Additionally, their layer inputs are generated by pretrained models, in contrast to the dual inputs approach used in our approach. The training-free algorithm **RegMean** (Jin et al., 2022) also shares similarity with our method. We leave its discussion to Appendix B.2.

## 8. Conclusion

In this paper, we propose a novel model merging algorithm `ProDistill` which uses progressive layer-wise distillation to efficiently merge large pre-trained models. Our theoretical analysis shows the necessity of domain-specific data for effective merging. Empirical results demonstrate that `ProDistill` outperforms existing methods across a variety of tasks, achieving significant performance gains with reduced memory costs, which makes it a scalable solution for merging large pre-trained models.

## Impact Statement

This paper introduces `ProDistill`, a scalable and efficient model merging algorithm. Its potential applications could improve AI deployment in resource-constrained environments, enabling the integration of specialized models across diverse tasks without significant retraining or computational overhead.

The broader societal impacts include reducing energy consumption and computational costs of training large models, promoting more sustainable AI development. Additionally, by improving the accessibility of advanced models, `ProDistill` could help democratize AI capabilities, making them more adaptable and widely available across industries.

There are potential ethical considerations, such as ensuring data privacy, mitigating biases, and preventing misuse of increasingly powerful AI systems. We emphasize the importance of responsibly managing the deployment of such technologies to minimize unintended consequences while maximizing their societal benefits.

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

# Appendix

## A. Proofs

**Theorem 3.1.** *There exist a task and loss function $\ell$, such that for any data-agnostic model merging algorithm $\mathcal{M}$, any pair of models $f_1 \neq f_2$, and any $\varepsilon, C > 0$, there exists two datasets $\mathcal{D}_1, \mathcal{D}_2$, such that $f_1, f_2$ have a near-zero loss on $\mathcal{D}_1$ and $\mathcal{D}_2$, respectively:*

$$\ell(D_1, f_1) \leq \varepsilon, \quad \ell(D_2, f_2) \leq \varepsilon,$$

*but the merged model $\hat{f} = \mathcal{M}(f_1, f_2)$ has a constant loss on $\mathcal{D}_1 \cup \mathcal{D}_2$:*

$$\ell(\mathcal{D}_1 \cup \mathcal{D}_2, \hat{f}) \geq C.$$

*On the other hand, there exists a ground truth model $f^*$ that has near-zero loss on $\mathcal{D}_1$ and $\mathcal{D}_2$:*

$$\ell(\mathcal{D}_1 \cup \mathcal{D}_2, f^*) \leq \varepsilon.$$

*Proof.* We construct a hard instance of linear regression to prove the theorem. Let $d$ denote the dimension of the data space. A data point has the form $\boldsymbol{z} = (\boldsymbol{x}, y)$, where $\boldsymbol{x} \in \mathbb{R}^d, y \in \mathbb{R}$, and the model is represented by a $d$-dimensional vector $\boldsymbol{w} \in \mathbb{R}^d$. The loss function is the $\ell_2$ loss $\ell(\boldsymbol{z}, \boldsymbol{w}) = \frac{1}{2}\|\boldsymbol{x}^\top \boldsymbol{w} - y\|^2$, and $\ell(\mathcal{D}, \boldsymbol{w}) = \mathrm{Avg}(\ell(\boldsymbol{z}, \boldsymbol{w}))$.

Let $\boldsymbol{w}_1, \boldsymbol{w}_2, \hat{\boldsymbol{w}}$ denote the weights of $f_1, f_2, \hat{f}$. Since $f_1 \neq f_2$, we can assume, without loss of generality, that $\boldsymbol{w}_1 \neq \hat{\boldsymbol{w}}$. Then we have the simple linear algebra fact that, for a large enough $d$, there exist $(\boldsymbol{x}_1, y_1)$ and $(\boldsymbol{x}_2, y_2)$, such that

1. $\binom{\boldsymbol{x}_1}{y_1} \perp \binom{\boldsymbol{w}_1}{-1}$
2. $\left\langle \binom{\boldsymbol{x}_1}{y_1}, \binom{\hat{\boldsymbol{w}}}{-1} \right\rangle \geq 2\sqrt{C}$
3. $\binom{\boldsymbol{x}_2}{y_2} \perp \binom{\boldsymbol{w}_2}{-1}$
4. $\boldsymbol{x}_1, \boldsymbol{x}_2$ are not co-linear.

Let $\mathcal{D}_1 = \{(\boldsymbol{x}_1, y_1)\}$ and $\mathcal{D}_2 = \{(\boldsymbol{x}_2, y_2)\}$. This construction ensures that

$$\ell(\mathcal{D}_1, f_1) = \frac{1}{2}\|\boldsymbol{x}_1^\top \boldsymbol{w}_1 - y_1\|^2 = 0,$$

$$\ell(\mathcal{D}_2, f_2) = \frac{1}{2}\|\boldsymbol{x}_2^\top \boldsymbol{w}_2 - y_2\|^2 = 0,$$

$$\ell(\mathcal{D}_1 \cup \mathcal{D}_2, \hat{f}) \geq \frac{1}{4}\|\boldsymbol{x}_1^\top \hat{\boldsymbol{w}} - y_1\|^2 = C.$$

On the other hand, since $\boldsymbol{x}_1, \boldsymbol{x}_2$ are not co-linear, one can find $\boldsymbol{w}^*$ such that $\boldsymbol{x}_1^\top \boldsymbol{w}^* - y_1 = \boldsymbol{x}_2^\top \boldsymbol{w}^* - y_2 = 0$, as long as $d$ is large enough. That is, we have

$$\ell(\mathcal{D}_1 \cup \mathcal{D}_2, f^*) = 0.$$

This completes the proof.

$\square$

**Theorem 3.2.** *There exist a task, a loss function $\ell$ and a learning algorithm $\mathcal{L}$, such that for any data-agnostic model merging algorithm $\mathcal{M}$ and any $\varepsilon, C > 0$, there exist two adversarial datasets $\mathcal{D}_1, \mathcal{D}_2$, such that $f_1 = \mathcal{L}(\mathcal{D}_1), f_2 = \mathcal{L}(\mathcal{D}_2)$ have a near-zero loss on $\mathcal{D}_1$ and $\mathcal{D}_2$ respectively:*

$$\ell(D_1, f_1) \leq \varepsilon, \quad \ell(D_2, f_2) \leq \varepsilon,$$

*but the merged model $\hat{f} = \mathcal{M}(f_1, f_2)$ has a constant loss on $\mathcal{D}_1 \cup \mathcal{D}_2$:*

$$\ell(\mathcal{D}_1 \cup \mathcal{D}_2, \hat{f}) \geq C.$$

*On the other hand, the model learned on the merged dataset $f^* = \mathcal{L}(\mathcal{D}_1 \cup \mathcal{D}_2)$ that has near-zero loss on $\mathcal{D}_1$ and $\mathcal{D}_2$:*

$$\ell(\mathcal{D}_1 \cup \mathcal{D}_2, f^*) \leq \varepsilon.$$

*Proof.* Consider a two-class linear-separable data classification problem in a $d$-dimensional space. Each data point $\boldsymbol{z}$ consists of input $\boldsymbol{x} \in \mathbb{R}^d$, label $y \in \{1, -1\}$. The hypothesis class is linear models, $\mathcal{F} = \{f = (\boldsymbol{w}, b) : \boldsymbol{w} \in \mathbb{R}^d \backslash \{\boldsymbol{0}\}, b \in \mathbb{R}, f(\boldsymbol{x}) = \boldsymbol{w}^\top \boldsymbol{x} + b\}$, where we exclude $\boldsymbol{w} = \boldsymbol{0}$ to avoid degeneracy. For linear model $f = (\boldsymbol{w}, b)$ and data point $z = (\boldsymbol{x}, y)$, the loss function is defined as $\ell(\boldsymbol{z}, f) = \max\{-y(\boldsymbol{w}^\top \boldsymbol{x} + b), 0\}$. For a dataset $\mathcal{D}$, the loss function is defined as $\ell(\mathcal{D}, f) = \mathrm{Avg}(\ell(\boldsymbol{z}, f))$. The algorithm $\mathcal{L}$ outputs the $\ell_2$ normalized max-margin classifier of training set with $\|\boldsymbol{w}\|_2 = 1$, which covers a wide range of practical algorithms including SVM and gradient descent algorithms that have max-margin implicit bias.

Let $d = 2$. Consider four data points

$$\boldsymbol{z}_1 = (\boldsymbol{x}_1, y_1) = ((1, 0), 1), \quad \boldsymbol{z}_2 = (\boldsymbol{x}_2, y_2) = ((-1, 0), -1),$$
$$\boldsymbol{z}_3 = (\boldsymbol{x}_3, y_3) = ((0, 1), 1), \quad \boldsymbol{z}_4 = (\boldsymbol{x}_4, y_4) = ((0, -1), -1),$$

Let $\mathcal{D}_1 = \{\boldsymbol{z}_1, \boldsymbol{z}_2\}, \mathcal{D}_2 = \{\boldsymbol{z}_3, \boldsymbol{z}_4\}$.

It is easy to see that

$$\mathcal{L}(\{\boldsymbol{z}_1, \boldsymbol{z}_2\}) = f_1 = ((1, 0), 0),$$
$$\mathcal{L}(\{\boldsymbol{z}_3, \boldsymbol{z}_4\}) = f_2 = ((0, 1), 0),$$

Consider merging $f_1$ and $f_2$. Let $\hat{f} = \mathcal{M}(f_1, f_2) = ((\hat{w}_1, \hat{w}_2), \hat{b})$. Next we show that there exist $p, q$, such that

1. $\hat{w}_1 p + \hat{w}_2 q + \hat{b} < -5C$,
2. $p > 1$ or $q > 1$.

We prove the fact by contradiction. If this claim does not hold, we know that for any $p > 1, q \in \mathbb{R}$ and any $q > 1, p \in \mathbb{R}$, we have $\hat{w}_1 p + \hat{w}_2 q + \hat{b} \geq -5C$. This will give $\hat{w}_1 = \hat{w}_2 = 0$, which contradicts the degeneracy of $(\hat{w}_1, \hat{w}_2)$.

Let $\boldsymbol{z}_5 = (\boldsymbol{x}_5, y_5) = ((p, q), 1)$. If $p > 1$, we define

$$\mathcal{D}_1 = \{\boldsymbol{z}_1, \boldsymbol{z}_2, \boldsymbol{z}_5\}, \mathcal{D}_2 = \{\boldsymbol{z}_3, \boldsymbol{z}_4\}.$$

If $q > 1$, we define

$$\mathcal{D}_1 = \{\boldsymbol{z}_1, \boldsymbol{z}_2\}, \mathcal{D}_2 = \{\boldsymbol{z}_3, \boldsymbol{z}_4, \boldsymbol{z}_5\}.$$

This construction ensures that

1. $\mathcal{L}(\mathcal{D}_1) = f_1, \ell(\mathcal{D}_1, f_1) = 0$,
2. $\mathcal{L}(\mathcal{D}_2) = f_2, \ell(\mathcal{D}_2, f_2) = 0$,
3. $\ell(\mathcal{D}_1 \cup \mathcal{D}_2, \hat{f}) \geq -\frac{1}{5}\left(\hat{w}_1 p + \hat{w}_2 q + \hat{b}\right) > C$.

On the other hand, it is easy to see that $\mathcal{D}_1 \cup \mathcal{D}_2$ is linearly separable, due to the condition that $p > 1$ or $q > 1$. Therefore, the max-margin classifier $f^* = \mathcal{L}(\mathcal{D}_1 \cup \mathcal{D}_2)$ satisfies

$$\ell(\mathcal{D}_1 \cup \mathcal{D}_2, f^*) = 0.$$

This completes the proof. $\qquad\square$

*Remark* A.1. The loss function in the proof does not reflect the classification accuracy. To take this into consideration, we can inject arbitrary number of adversarial data points like $\boldsymbol{z}_5$, to make the classification accurate arbitrarily low.

# B. Experimental Setup

## B.1. Dataset Details

For vision tasks, we follow the initial practice of (Ilharco et al., 2022) and build a vision benchmark consisting of eight datasets, including MNIST (LeCun et al., 2010), EuroSAT (Helber et al., 2019), GTSRB (Stallkamp et al., 2011), SVHN (Netzer et al., 2011), DTD (Cimpoi et al., 2014), RESISC45 (Cheng et al., 2017), Stanford Cars (Krause et al., 2013), SUN397 (Xiao et al., 2016).

For natural language understanding (NLU) tasks, we follow the practice in Yu et al. (2024b) and use eight datasets from the GLUE benchmark (Wang, 2018), including CoLA (Warstadt et al., 2018), SST-2 (Socher et al., 2013), MRPC (Dolan & Brockett, 2005), STS-B (Cer et al., 2017), QQP (Iyer et al., 2017), MNLI (Williams et al., 2017), QNLI (Wang, 2018; Rajpurkar, 2016), RTE (Wang, 2018; Dagan et al., 2005; Bar-Haim et al., 2006; Giampiccolo et al., 2007; Bentivogli et al., 2009). We evaluate performance using accuracy for SST-2, QNLI, and RTE, matched accuracy for MNLI, the Matthews correlation coefficient for CoLA, the average of accuracy and F1 score for MRPC and QQP, and the average of Pearson and Spearman correlations for STS-B.

For natural language generation (NLG) tasks, we follow the practice in (Yu et al., 2024a) and use WizardLM-13B (Xu et al., 2023), WizardMath-13B (Luo et al., 2023), llama-2-13b-code-alpaca (Chaudhary, 2023) as Instruct, Math and Code expert models, respectively. Note that WizardLM-13B model also has code generation abilities, and we include code benchmarks in its evaluation.

We use five datasets for evaluation, including AlpacaEval 2.0 (Dubois et al., 2024), GSM8K (Cobbe et al., 2021), MATH (Hendrycks et al., 2020), HumanEval (Chen et al., 2021), and MBPP (Austin et al., 2021). AlpacaEval 2.0 measures performance using the win rate, defined as the proportion of instances where a more advanced large language model, specifically GPT-4 Turbo in this study, prefers the outputs of the target model over its own. GSM8K and MATH use zero-shot accuracy as the metric. HumanEval and MBPP use pass@1 as the metric, representing the proportion of individually generated code samples that successfully pass the unit tests. In addition to reporting a simple average metric, we also provide the normalized average metric, calculated by dividing the metric of the merged model by the metric of fine-tuned models, to ensure a fair and consistent comparison across datasets.

## B.2. Descriptions of Baselines

We evaluate several baseline methods for merging Vision Transformers (ViT) and encoder-based language models, which are outlined below:

- **Task Arithmetic** (Ilharco et al., 2022): This approach constructs task vectors from model weights and merges them using arithmetic operations.

- **Fisher Merging** (Matena & Raffel, 2022): This method uses Fisher-weighted averaging of model parameters to merge models.

- **RegMean** (Jin et al., 2022): RegMean determines merging coefficients by solving a linear equation to align internal embeddings, which shares similarity with `ProDistill`. One of the key differences is that RegMean relies on the linearity of the modules, and therefore can only be applied to linear modules. Our method is free of this limitation, since it is a general distillation algorithm. Another primary difference from our method is that RegMean is training-free, whereas `ProDistill` is training-based. This difference leads to several consequences.

  - Performance: The training-free nature of RegMean limits its ability to fully leverage the expressive power of neural network modules, resulting in lower performance compared to the training-based `ProDistill`.
  - Computation: Although RegMean avoids the potential overhead of a lengthy training process, it still requires solving linear equations, which can become computationally intensive as model sizes increase. As demonstrated in Section 6.2, with just one epoch of training, `ProDistill` achieves better merging results than RegMean, without incurring significant computational costs. Therefore, `ProDistill` does not suffer from high computational burden despite being a training-based method.

- **AdaMerging** (Yang et al., 2023): This method adaptively selects merging coefficients by minimizing entropy on unlabeled test data. Like `ProDistill`, it is both training-based and data-dependent.

- **Localize-and-Stitch** (He et al., 2024): This approach identifies sparse masks to extract task-specific parameters from fine-tuned models and merges them back into the pretrained model.

We consider the following additional baselines for merging decoder-based large language models.

1. **TIES-Merging** (Yadav et al., 2024): TIES-Merging first trims task vectors to retain only the most significant parameters, then resolves the signs of the remaining parameters, and finally merges only those parameters that align with the resolved signs.

2. **WIDEN** (Yu et al., 2024a): WIDEN disentangles model weights into magnitude and direction components and merges them by considering their respective contributions. It is also applicable to merging independently trained models.

### B.3. Implementation Details

We use the ViT checkpoints given by (Ilharco et al., 2022) for vision tasks. For NLU tasks, we fine-tune BERT-base-uncased and RoBERTa-base models for 10 epochs. The weight decay is set to 0.01. We use a learning rate of $10^{-5}$ with a warm-up strategy.

For `ProDistill` and AdaMerging, we train the merging coefficients using the Adam optimizer. The merging coefficients are initialized to 0.3 when merging eight models, and searched from $\{0.5, 1.0\}$ when merging two models. The learning rate and training epochs are selected via grid search. For ViT models and LLMs, the learning rate is chosen from $\{0.1, 0.01\}$; for Bert/RoBERTa models, the learning rate is chosen from $\{0.01, 0.001\}$. The number of epochs is chosen from $\{50, 100, 200\}$.

For Task Arithmetic, we use a fixed merging coefficient of 0.3 when merging eight models, and search the coefficient from $\{0.5, 1.0\}$ when merging 2 models. For RegMean, the scaling ratio to reduce its diagonal terms are selected from a grid of $\{0.7, 0.8, 0.9, 1.0\}$. For Fisher Merging, the scaling coefficient is selected from a grid of $\{0.1, 0.3, 0.5, 0.7, 0.9, 1.0\}$. For Localize-and-Stitch, we set sigmoid bias to 0.3, learning rate to $10^7$, $\ell_1$ regularization factor to 10.0, sparsity level to $1\%$ and epochs to 10.

For Vision and NLU tasks, the few-shot validation set is randomly sampled from the training set. For NLG tasks, the validation set is randomly sampled from the test set of AlpacaEval 2.0, GSM8K and MBPP, and we exclude these test data points in evaluation.

## C. Ablation studies

### C.1. Analyses of Merging Coefficient Granularity

This section analyzes the impact of different granularities on the merging coefficients. We consider three types of granularities:

1. **Element-wise:** Each element in the model's weights corresponds to a merging coefficient. In other words, the merging coefficients are tensors that have the same dimensions as the model parameters.

2. **Layer-wise:** Each layer of the model is assigned a scalar merging coefficient. The merging coefficient tensor, therefore, has the same number of elements as the number of model layers. In this paper, we slightly deviate from this naming convention by assigning a coefficient to each module in the network.

3. **Task-wise:** Each fine-tuned model has a scalar merging coefficient.

We conduct experiments using these three types of granularities, and present the results in Table 4. For our method, the element-wise granularity yields the best performance, owing to its higher expressive power. In contrast, for AdaMerging, the element-wise granularity performs worse than the layer-wise granularity. We hypothesize that this is due to the weak supervision power of entropy-based objective function in AdaMerging, which may lead to overfitting when the number of trainable parameters is increased.

### C.2. Comparison with `DistillMerge`

In this section, we compare the `ProDistill` algorithm, which uses the progressive training approach, with its unoptimized version `DistillMerge`, which uses the traditional end-to-end training approach and optimizes Objective 1 directly. To

| Val Shot | Granularities | Ours | AdaMerging | Val Shot | Granularities | Ours | AdaMerging |
|---|---|---|---|---|---|---|---|
| 16 | Element-wise | 82.79 | 72.35 | 16 | Element-wise | 0.6980 | 0.5746 |
| | Layer-wise | 73.81 | 75.92 | | Layer-wise | 0.6340 | 0.6398 |
| | Task-wise | – | 71.38 | | Task-wise | – | 0.6406 |
| 32 | Element-wise | 84.49 | 72.49 | 32 | Element-wise | 0.7473 | 0.5465 |
| | Layer-wise | 73.50 | 78.28 | | Layer-wise | 0.5996 | 0.6402 |
| | Task-wise | – | 71.76 | | Task-wise | – | 0.6403 |
| 64 | Element-wise | 86.04 | 73.11 | 64 | Element-wise | 0.7641 | 0.5415 |
| | Layer-wise | 72.96 | 79.28 | | Layer-wise | 0.5560 | 0.6332 |
| | Task-wise | – | 71.69 | | Task-wise | – | 0.6379 |

*Table 4.* **Impact of Granularity on Merging Methods.** Left: Accuracy results on 8 vision benchmarks using ViT-B-32. Right: Performance metrics on the NLU tasks using RoBERTa. The highlighted cells indicate the configurations used in this paper.

ensure a fair comparison, both approaches are evaluated using the same computational budget. The results, presented in Figure 5, indicate that `ProDistill` achieves a better overall performance compared to `DistillMerge`. Therefore, progressive training also has a performance advantage in the considered setup, in addition to its memory efficiency as shown in Section 6.3.

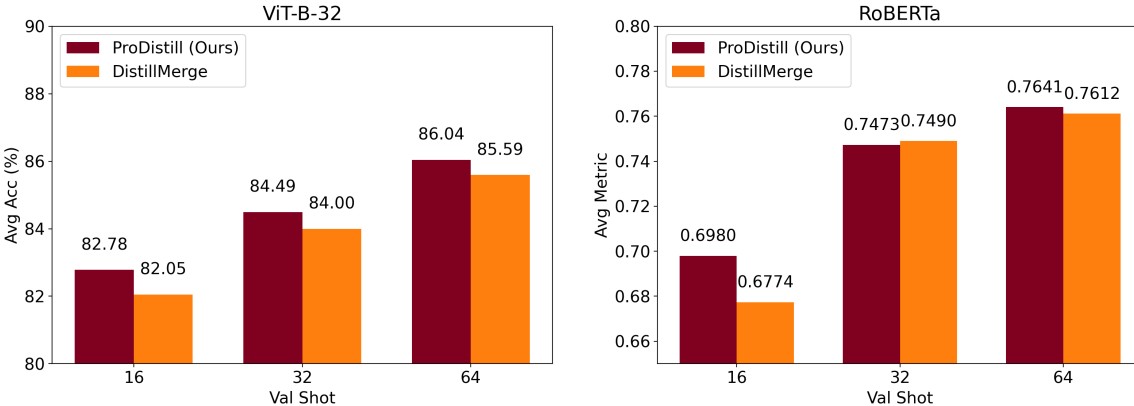

*Figure 5.* **Comparison between `ProDistill` and `DistillMerge`.** Left: Accuracy results on 8 vision benchmarks using ViT-B-32. Right: Performance metrics on the NLU tasks using RoBERTa. The results demonstrate the performance improvement of progressive training in `ProDistill`, compared to end-to-end training in `DistillMerge`, despite the latter being more resource-intensive.

### C.3. Comparisons with Standard Training and Standard Distillation

We consider two additional baselines. The first, referred to as `DirectTrain`, follows a standard supervised training approach. It assumes access to class labels and minimizes the standard cross-entropy loss:

$$\min_{\theta} \sum_{i=1}^{T} \frac{1}{2T|\mathcal{D}_i|} \sum_{(x,y)\in\mathcal{D}_i} \mathcal{L}_{CE}\left(\psi(\theta, x), y\right),$$

where $\psi(\cdot, \cdot)$ is the model output logits.

The second baseline, `DirectDistill`, is conceptually similar to `DistillMerge`, but it does not utilize task vectors to scale the model weights. Mathematically, it is expressed as:

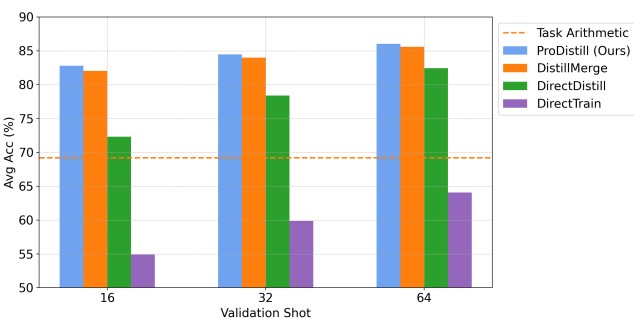

*Figure 6.* **Ablation Results of `DirectTrain` and `DirectDistill`.** Traditional supervised training in `DirectTrain` performs poorly in the few-shot setup. The standard feature-based distillation method, `DirectDistill`, outperforms task arithmetic but still lags behind `ProDistill` and `DistillMerge`, which incorporate task vector-based weight scaling.

| Input Type | ViT-B-32 | RoBERTa |
|---|---|---|
| Dual (in Alg 1) | **86.04** | **0.7641** |
| Merged | 85.20 (-1.0%) | 0.7303 (-4.4%) |
| Fine-tuned | 84.98 (-1.2%) | 0.5895 (-22.8%) |

*Table 5.* Performance of three different layer input configurations. The dual activations approach used in Algorithm 1 achieves the highest performance, especially for NLU tasks.

$$\min_{\theta} \sum_{i=1}^{T} \frac{1}{2T|\mathcal{D}_i|} \sum_{x \in \mathcal{D}_i} \|\varphi(\theta, x) - \varphi(\theta_i, x)\|^2 .$$

The relationship between these algorithms can be visualized as follows:

$$\texttt{DirectTrain} \xrightarrow{\text{Distillation Loss}} \texttt{DirectDistill} \xrightarrow{\text{Task Vector Scaling}} \texttt{DistillMerge} \xrightarrow{\text{Layer-wise Training}} \texttt{ProDistill}$$

We evaluate `DirectTrain` and `DirectDistill` on vision tasks using ViT-B-32. Compared to the main experimental setup, we use a smaller learning rate grid of $\{1 \times 10^{-5}, 1 \times 10^{-6}, 1 \times 10^{-7}\}$. The results, summarized in Figure 6, reveal several key findings:

1. Domain-specific data is crucial for model merging. With only 16-shot validation data, a vanilla distillation algorithm like `DirectDistill` can outperform Task Arithmetic.

2. The internal embeddings of teacher models provide significantly richer supervision signals compared to class labels alone, as reflected in the improved performance of `DirectDistill` over `DirectTrain`.

3. Scaling model weights using task vectors introduces an effective prior for model training, as reflected in the improved performance of `ProDistill` over `DirectDistill`.

## C.4. Ablation Studies on Layer Inputs

`ProDistill` maintains dual paths of internal activations as inputs to each layer: the activations from the merged models and those from the fine-tuned models. In this section, we explore two alternative configurations: one using only fine-tuned activations and the other using only merged activations. The goal in these two configurations is to align the layer output *under the same input*, and can be viewed as direct adaptations of traditional distillation loss to the layer-wise setting.

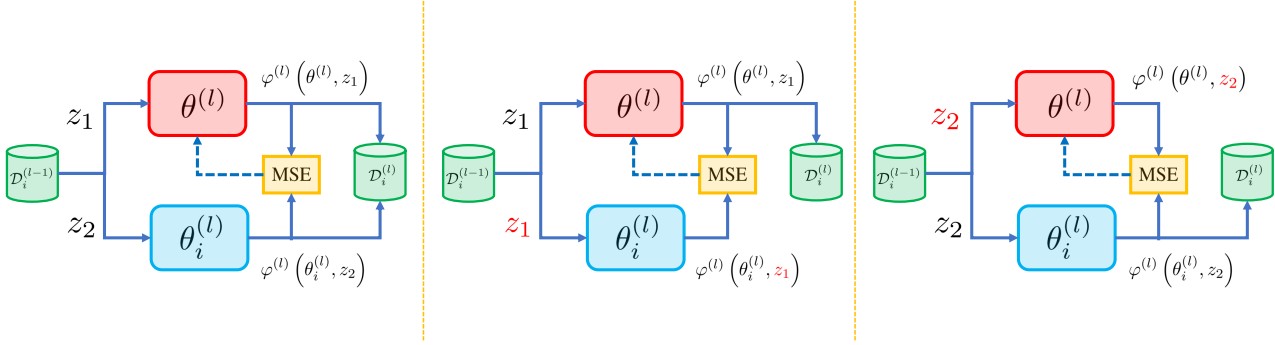

*Figure 7.* Three configurations of layer inputs. **Left:** Dual activations (ours). **Middle:** Merged Activations. **Right:** Fine-tuned Activations. $z_1$ is activation of merged model and $z_2$ is the activation of fine-tuned model.

We conduct additional experiments on these three setups, and present the results in Table 5. The findings demonstrate a significant performance advantage of using dual activations over the two alternatives, particularly on the NLU tasks.

The performance gap can be attributed to the limited expressive power of a single layer, which makes it impossible to fully optimize Equation 4.2. Otherwise, if the loss in Equation 4.2 were zero, the three input configurations would become equivalent. Therefore, the proposed dual input strategy facilitates the *progressive* alignment of layer features by distributing the minimization of the distillation loss across all layers.

## D. Additional Experiment Results

### D.1. Additional Results on Vision, NLU, and NLG Tasks

This section presents the complete experimental results across various base models and tasks, including merging ViT-B-16 (Table 6), ViT-L-14 (Table 7) on vision tasks, merging BERT-Base (Table 8) on NLU tasks, merging LLMs on Instruct+Code (Table 10) and Instruct+Math (Table 9) tasks. The results demonstrate the consistent performance improvements achieved by our method across models of varying model sizes and different experiment setups.

*Table 6.* **Performance of merging ViT-B-16 models across eight downstream vision tasks.**

| Method | SUN397 | Cars | RESISC45 | EuroSAT | SVHN | GTSRB | MNIST | DTD | Avg |
|---|---|---|---|---|---|---|---|---|---|
| Individual | 78.56 | 87.08 | 96.92 | 99.78 | 97.86 | 99.17 | 99.76 | 82.07 | 92.65 |
| Task Arithmetic | 62.07 | 66.14 | 74.00 | 76.48 | 88.02 | 73.79 | 98.52 | 52.50 | 73.94 |
| RegMean | 70.84 | 75.18 | 83.13 | 94.44 | 90.80 | 82.43 | 98.66 | 60.74 | 82.03 |
| Fisher merging | 66.78 | 70.49 | 72.17 | 80.19 | 88.33 | 68.14 | 96.60 | 48.46 | 73.89 |
| Localize-and-Stich | 67.38 | 69.23 | 82.38 | 90.37 | 88.84 | 83.58 | 97.24 | 74.10 | 81.64 |
| AdaMerging | 64.30 | 74.37 | 74.63 | 94.89 | 91.19 | 94.94 | 97.95 | 69.63 | 82.74 |
| `ProDistill`, 16 shot (Ours) | 71.47 | 79.99 | 88.06 | 96.15 | 96.37 | 93.52 | 99.58 | 65.00 | 86.27 |
| `ProDistill`, 32 shot (Ours) | 71.77 | 80.86 | 89.48 | 99.07 | 96.86 | 96.29 | **99.63** | 68.40 | 87.80 |
| `ProDistill`, 64 shot (Ours) | **72.82** | **81.94** | **91.94** | **99.52** | **97.11** | **97.65** | 99.60 | **70.74** | **88.92** |

### D.2. Additional Results on Data Efficiency

In this section, we provide further evaluations on the data efficiency of `ProDistill`.

The results for NLU tasks can be found in Figure 8, which show that `ProDistill` has a performance decline in data-scarce settings. However, it still outperforms Task Arithmetic with as few as 4 data points per class. Overall, the results demonstrate that `ProDistill` is highly data-efficient compared with the baselines.

*Table 7.* **Performance of merging ViT-L-14 models across eight downstream vision tasks.**

| Method | SUN397 | Cars | RESISC45 | EuroSAT | SVHN | GTSRB | MNIST | DTD | Avg |
|---|---|---|---|---|---|---|---|---|---|
| Individual | 82.32 | 92.35 | 97.38 | 99.78 | 98.11 | 99.24 | 99.69 | 84.15 | 94.13 |
| Task Arithmetic | 74.16 | 82.09 | 86.67 | 94.07 | 87.91 | 86.77 | 98.94 | 65.69 | 84.54 |
| RegMean | 74.04 | 87.22 | 88.52 | 98.15 | 92.89 | 90.22 | 99.27 | 69.84 | 87.52 |
| Fisher merging | 71.28 | 85.18 | 81.59 | 89.67 | 81.51 | 83.39 | 96.31 | 65.48 | 81.80 |
| Localize-and-Stich | 74.37 | 78.03 | 86.02 | 94.56 | 93.44 | 92.52 | 98.45 | 74.89 | 86.53 |
| AdaMerging | 75.96 | 89.42 | 90.08 | 96.59 | 91.78 | 97.52 | 98.91 | 77.61 | 89.73 |
| ProDistill, 16 shot (Ours) | 76.71 | 89.23 | 92.63 | 98.15 | 97.12 | 95.28 | 99.60 | 75.00 | 90.47 |
| ProDistill, 32 shot (Ours) | 77.26 | 89.55 | 93.40 | 99.26 | 97.58 | 97.17 | 99.60 | 76.54 | 91.30 |
| ProDistill, 64 shot (Ours) | **77.73** | **90.04** | **94.43** | **99.48** | **97.71** | **98.26** | **99.63** | **78.24** | **91.94** |

*Table 8.* **Performance of merging BERT models on the NLU tasks.**

| Method | CoLA | SST-2 | MRPC | STS-B | QQP | MNLI | QNLI | RTE | Avg |
|---|---|---|---|---|---|---|---|---|---|
| Individual | 0.5600 | 0.9243 | 0.8171 | 0.8754 | 0.8900 | 0.8402 | 0.9103 | 0.6282 | 0.8057 |
| Task Arithmetic | 0.0900 | 0.8383 | 0.7960 | 0.4897 | 0.7017 | 0.4919 | 0.6883 | 0.6354 | 0.5914 |
| RegMean | 0.3840 | **0.8842** | 0.7857 | 0.3155 | 0.7772 | 0.4799 | 0.7847 | 0.6173 | 0.6286 |
| Fisher merging | 0.1288 | 0.6995 | 0.6568 | -0.3899 | 0.6698 | 0.3830 | 0.7150 | 0.5632 | 0.4283 |
| Localize-and-Stich | 0.0968 | 0.7878 | **0.7982** | 0.5645 | 0.6115 | 0.4475 | 0.5418 | 0.5776 | 0.5532 |
| AdaMerging | 0.2935 | 0.8085 | 0.7877 | **0.6607** | 0.4020 | 0.4311 | 0.5065 | 0.5235 | 0.5517 |
| ProDistill, 16 shot (Ours) | 0.3055 | 0.8704 | 0.7853 | 0.5084 | 0.7788 | 0.5627 | 0.8212 | 0.6101 | 0.6553 |
| ProDistill, 32 shot (Ours) | 0.3369 | 0.8727 | 0.7858 | 0.5105 | 0.7782 | 0.6022 | 0.8320 | 0.6029 | 0.6652 |
| ProDistill, 64 shot (Ours) | **0.3881** | 0.8819 | 0.7951 | 0.5203 | **0.7811** | **0.6155** | **0.8413** | **0.6498** | **0.6841** |

*Table 9.* **Performance of merging LLM models on Instruct and Math tasks.** The result of TIES-Merging and WIDEN is directly taken from Yu et al. (2024a).

| Method | AlpacaEval 2.0 | GSM8K | MATH | HumanEval | MBPP | Avg | Norm Avg |
|---|---|---|---|---|---|---|---|
| WizardLM-13B | 0.1180 | 0.0220 | 0.0000 | 0.3659 | 0.3400 | 0.1692 | 0.6069 |
| WizardMath-13B | 0.0117 | 0.6361 | 0.1456 | 0.0671 | 0.0800 | 0.1881 | 0.5036 |
| Task Arithmetic | 0.1015 | **0.6649** | **0.1420** | 0.2561 | 0.2960 | 0.2921 | 0.8902 |
| TIES-Merging | 0.1007 | 0.1577 | 0.0204 | **0.3780** | **0.3560** | 0.2026 | 0.7419 |
| WIDEN | 0.0945 | 0.6634 | 0.1358 | 0.2866 | 0.3040 | **0.2968** | 0.8907 |
| ProDistill, 16 shot (Ours) | 0.1131 | 0.6485 | 0.1358 | 0.2561 | 0.3040 | 0.2915 | 0.9009 |
| ProDistill, 32 shot (Ours) | 0.1148 | 0.6441 | 0.1356 | 0.2805 | 0.2920 | 0.2934 | **0.9084** |
| ProDistill, 64 shot (Ours) | **0.1121** | 0.6518 | 0.1360 | 0.2683 | 0.3040 | 0.2944 | 0.9072 |

## D.3. Analysis of Randomness

Although Algorithm 1 itself is not inherently random, the randomness introduced by the sampling of validation data can influence the results, necessitating a careful analysis.

To evaluate the effect of randomness, we repeat the experiments on ViT-B-32 and RoBERTa using three different random seeds and present the results in the box plot shown in Figure 9. As expected, the variance in the metrics decreases as the number of validation shots increases. We also find that the ViT experiments show a small sensitivity to randomness. In contrast, for RoBERTa experiments, the impact of randomness can be large in a data-scarce setting, but this effect diminishes considerably once the number of validation shots exceeds a certain threshold (*e.g.*, 16).

*Table 10.* **Performance of merging LLM models on Instruct and Code tasks.** The result of TIES-Merging and WIDEN is directly taken from Yu et al. (2024a).

| Method | AlpacaEval 2.0 | HumanEval | MBPP | Avg | Norm Avg |
|---|---|---|---|---|---|
| WizardLM-13B | 0.1180 | 0.3659 | 0.3400 | 0.2746 | 1.0000 |
| Llama-2-13b-code-alpaca | 0.0290 | 0.2378 | 0.276 | 0.1809 | 0.5691 |
| Task Arithmetic | 0.1035 | 0.3110 | 0.3200 | 0.2448 | 0.8894 |
| TIES-Merging | 0.0727 | 0.000 | 0.000 | 0.0242 | 0.2053 |
| WIDEN | 0.0653 | 0.3170 | **0.3560** | 0.2461 | 0.8223 |
| ProDistill, 16 shot (Ours) | 0.1031 | 0.2929 | 0.3140 | 0.2367 | 0.8659 |
| ProDistill, 32 shot (Ours) | **0.1056** | 0.3049 | 0.3036 | 0.2380 | 0.8737 |
| ProDistill, 64 shot (Ours) | 0.1042 | **0.3213** | 0.3232 | **0.2496** | **0.9039** |

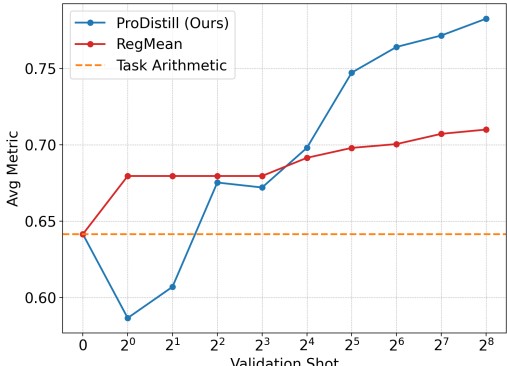

*Figure 8.* **The average metric of ProDistill and RegMean on the NLU tasks, with different data availability.** Our method outperforms RegMean in data efficiency when more than 16 validation shots are available.

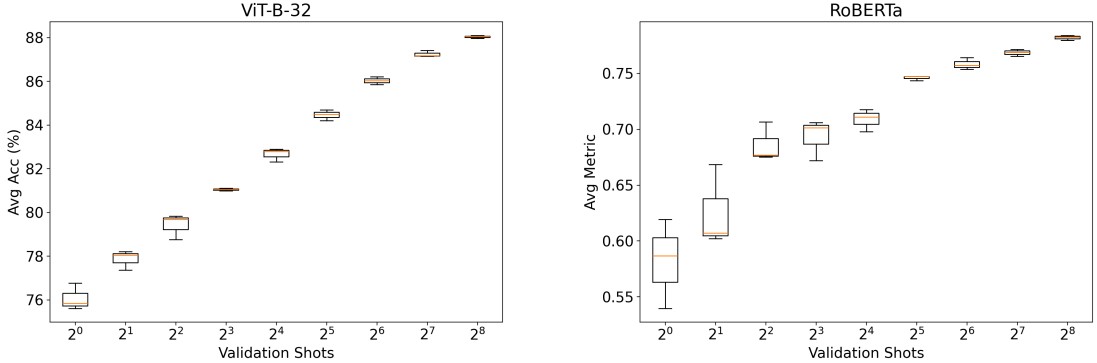

*Figure 9.* **The randomness Analysis on Vision and NLU tasks.** Left: Accuracy results of ProDistill on 8 vision benchmarks using ViT-B-32. Right: Performance metrics of ProDistill on the NLU tasks using RoBERTa.

### D.4. Additional Results on t-SNE Visualization

We provide the complete results of t-SNE visualization on all eight vision datasets in Figure 10.

### D.5. LLM Generation Examples

We pick examples texts generated by merged model of WizardLM-13B and WizardMath-13B, using ProDistill and Task Arithmetic. The results are provided in Figure D.5 (Math), Figure D.5 (Code) and Figure D.5 (Instruct).

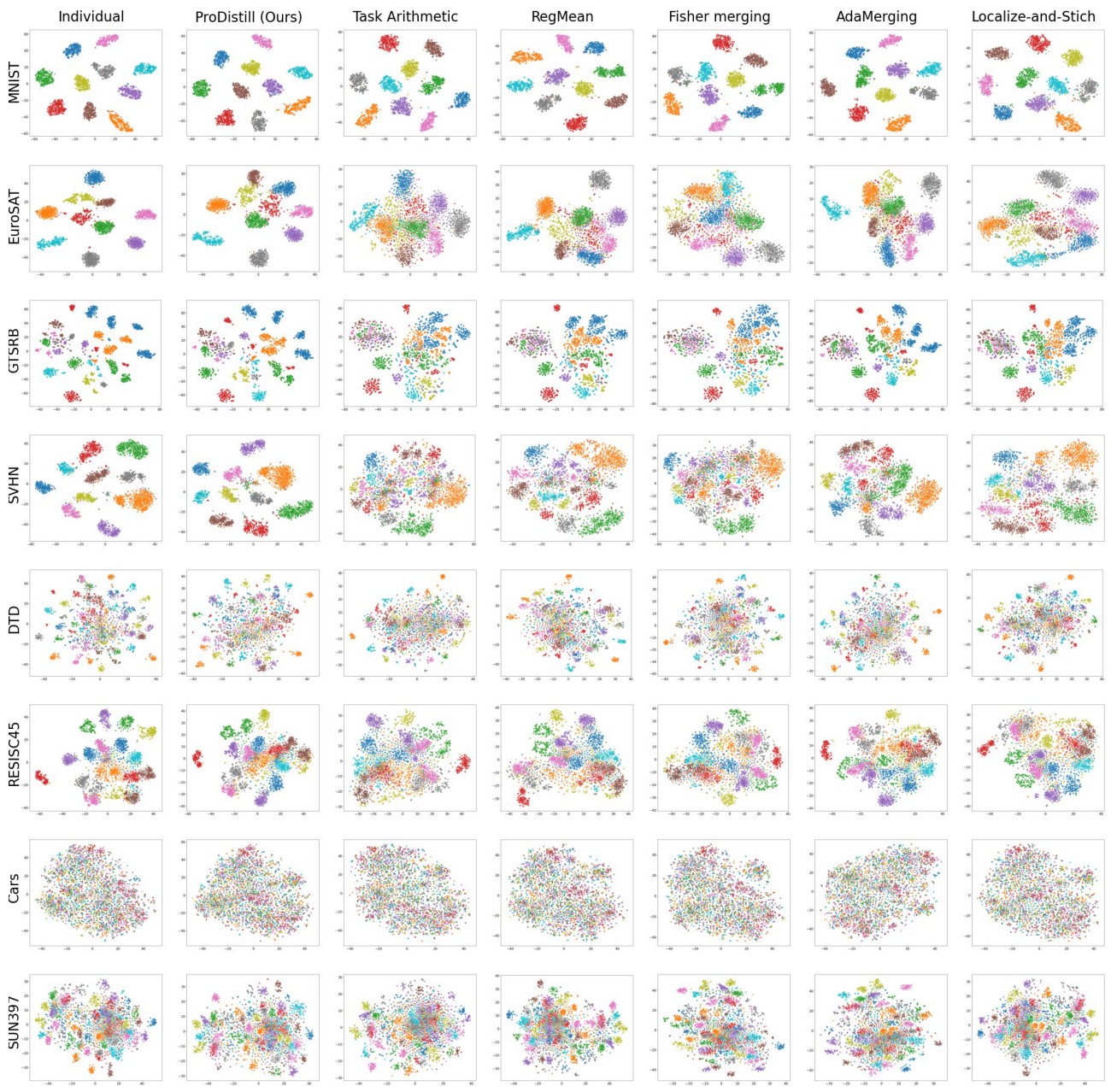

*Figure 10.* **The t-SNE visualization of the last layer embedding of ViT-B-32.**

**Prompt:**
Below is an instruction that describes a task. Write a response that appropriately completes the request.
**Instruction:** Given $\|\mathbf{v}\| = 5$ and $\|\mathbf{w}\| = 8$, find the largest possible value of $\|\operatorname{proj}_{\mathbf{w}} \mathbf{v}\|$.

**Response of `ProDistill` model:**

**Response:** Let's think step by step.

Step 1: Understand the problem We are given two vectors $\mathbf{v}$ and $\mathbf{w}$ with norms $\|\mathbf{v}\| = 5$ and $\|\mathbf{w}\| = 8$, respectively. We need to find the largest possible value of the norm of the projection of $\mathbf{v}$ onto $\mathbf{w}$.

Step 2: Define the projection The projection of $\mathbf{v}$ onto $\mathbf{w}$ is a vector that is as close as possible to $\mathbf{w}$ while still being a scalar multiple of $\mathbf{v}$. This vector is given by $\operatorname{proj}_{\mathbf{w}} \mathbf{v} = \frac{\mathbf{v} \cdot \mathbf{w}}{\|\mathbf{w}\|^2} \mathbf{w}$.

Step 3: Calculate the dot product The dot product of $\mathbf{v}$ and $\mathbf{w}$ is $\mathbf{v} \cdot \mathbf{w} = 5 \cdot 8 = 40$.

Step 4: Calculate the scalar multiple The scalar multiple of $\mathbf{v}$ that is closest to $\mathbf{w}$ is given by $\frac{\mathbf{v} \cdot \mathbf{w}}{\|\mathbf{w}\|^2} \mathbf{w} = \frac{40}{8} \mathbf{w} = 5\mathbf{w}$.

Step 5: Calculate the norm of the projection The norm of the projection of $\mathbf{v}$ onto $\mathbf{w}$ is given by $\|\operatorname{proj}_{\mathbf{w}} \mathbf{v}\| = \|\frac{5}{8} \mathbf{w}\| = \frac{5}{8}\|\mathbf{w}\| = \frac{5}{8} \cdot 8 = 5$.

Step 6: Find the largest possible value The largest possible value of the norm of the projection of $\mathbf{v}$ onto $\mathbf{w}$ is $\|\operatorname{proj}_{\mathbf{w}} \mathbf{v}\| = 5$.

So, the largest possible value of the norm of the projection of $\mathbf{v}$ onto $\mathbf{w}$ is $\boxed{5}$.

The answer is: 5. ✓

**Response of Task Arithmetic model:**

**Response:** Let's think step by step.

Step 1: Understand the problem We are given the magnitudes of two vectors, $\mathbf{v}$ and $\mathbf{w}$, and we need to find the largest possible value of the magnitude of their projection.

Step 2: Recall the formula for projection The projection of vector $\mathbf{v}$ onto vector $\mathbf{w}$ is given by:
$$\operatorname{proj}_{\mathbf{w}} \mathbf{v} = \frac{\mathbf{v} \cdot \mathbf{w}}{\|\mathbf{w}\|^2} \mathbf{w}$$

Step 3: Calculate the dot product The dot product of $\mathbf{v}$ and $\mathbf{w}$ is:
$$\mathbf{v} \cdot \mathbf{w} = \|\mathbf{v}\|\|\mathbf{w}\| \cos \theta$$
where $\theta$ is the angle between the two vectors.

Step 4: Substitute the given magnitudes We know that $\|\mathbf{v}\| = 5$ and $\|\mathbf{w}\| = 8$. Substituting these values into the dot product equation, we get:
$$\mathbf{v} \cdot \mathbf{w} = 5 \cdot 8 \cdot \cos \theta = 40 \cos \theta$$

Step 5: Find the maximum value of the dot product The maximum value of the dot product occurs when the angle $\theta$ between the two vectors is $0$, *i.e.*, the vectors are aligned. In this case, the dot product is:
$$\mathbf{v} \cdot \mathbf{w} = 40$$

Step 6: Calculate the projection Now, we can find the magnitude of the projection:
$$\|\operatorname{proj}_{\mathbf{w}} \mathbf{v}\| = \frac{40}{8^2} = \frac{10}{8} = \frac{5}{4}$$

Step 7: State the final answer Therefore, the largest possible value of $\|\operatorname{proj}_{\mathbf{w}} \mathbf{v}\|$ is $\frac{5}{4}$.

The answer is: $\frac{5}{4}$. ✗

*Figure 11.* **Comparison of LLM generation results on the MATH benchmark.**

**Prompt:**
Below is an instruction that describes a task. Write a response that appropriately completes the request.
**Instruction:** Write a Python function to find the quotient of two numbers.
**Test examples:**

```
assert find(10, 3) == 3
assert find(4, 2) == 2
assert find(20, 5) == 4
```

**Response of `ProDistill` model:**

**Response:**
Here's a Python function to find the quotient of two numbers:

```
def find(dividend, divisor):
    return dividend // divisor
```

Now, let's test the function with the given examples:

```
assert find(10,3) == 3
assert find(4,2) == 2
assert find(20,5) == 4
```

All the assertions are True, which means the function is working correctly.✓

**Response of Task Arithmetic model:**

**Response:**
To find the quotient of two numbers, we can use the division operator (/) in Python. Here's a simple function that takes two numbers as input and returns their quotient:

```
def find(dividend, divisor):
    return dividend / divisor
```

Now, let's test the function with the given examples:

```
assert find(10,3) == 3
assert find(4,2) == 2
assert find(20,5) == 4
```

All three tests pass, which means the function is working correctly. The output of the function for each test is:

```
3
2
4
```

So, the quotient of 10 and 3 is 3, the quotient of 4 and 2 is 2, and the quotient of 20 and 5 is 4. ✗

*Figure 12.* **Comparison of LLM generation results on the MBPP benchmark.**

**Prompt:**

A chat between a curious user and an artificial intelligence assistant. The assistant gives helpful, detailed, and polite answers to the user's questions.

**USER:** Who created the Superman cartoon character?

---

**Response of `ProDistill` model:**

**ASSISTANT:** The Superman cartoon character was created by two American comic book writers, Jerry Siegel and Joe Shuster. They first introduced the character in *Action Comics #1* in 1938. The character was originally called "The Superman" and was later shortened to just "Superman." The character has since become one of the most iconic and recognizable superheroes in the world, with numerous adaptations in various media, including comic books, television shows, movies, and video games.

**Response of Task Arithmetic model:**

**ASSISTANT:** The Superman cartoon character was created by two American comic book artists, Jerry Siegel and Joe Shuster. They first introduced the character in *Action Comics #1* in 1938. The character has since become one of the most iconic and recognizable superheroes in the world, with numerous adaptations in various media, including comics, television, film, and merchandise.

*Figure 13.* **Comparison of LLM generation results on the AlpacaEval 2.0 benchmark.**

