# OpenReview forum: "Scalable Model Merging with Progressive Layer-wise Distillation"
_ICML.cc/2025/Conference — ICML 2025 poster_

### Official Review · Reviewer_iWn3 · 2025-02-27

**Overall Recommendation:** 3

**Summary:**

The paper introduces ​ProDistill, a ​progressive layer-wise distillation​ algorithm for merging multiple fine-tuned models into a single high-performing model. It theoretically demonstrates the necessity of ​task-specific data​ for effective merging and proposes a layer-by-layer distillation approach that minimizes activation distance between teacher (fine-tuned) and student (merged) models, reducing memory overhead.

**Claims And Evidence:**

yes

**Essential References Not Discussed:**

No

**Ethics Expertise Needed:**

["Other expertise"]

**Experimental Designs Or Analyses:**

yes

**Methods And Evaluation Criteria:**

yes

**Other Comments Or Suggestions:**

No

**Other Strengths And Weaknesses:**

It introduces a new method called ​ProDistill, which uses ​layer-wise distillation​ to merge models, addressing the performance drop issue in existing methods. The theoretical analysis on the necessity of task-specific data is convincing.
It works well with ​limited data​ (e.g., 1-shot) and scales to ​10+ billion parameters, making it suitable for real-world applications.

Experiments are limited to vision and NLP tasks, with no validation in other domains like speech or reinforcement learning.

**Questions For Authors:**

The experiments focus on vision and NLP tasks. Have you tested ProDistill on other domains, such as speech or reinforcement learning? If not, do you believe the method would generalize well to these areas?

Will you release the code and implementation details for ProDistill? If not, what are the reasons?

While layer-wise distillation reduces memory overhead, are there any trade-offs in terms of performance or training stability compared to end-to-end distillation?

**Relation To Broader Scientific Literature:**

The key contributions of the paper are closely tied to and build upon ​broader scientific literature​ in model merging, knowledge distillation, and multi-task learning

**Theoretical Claims:**

yes

---

> ### Author Rebuttal · Authors · 2025-03-31
>
> We would like to express our sincere gratitude to the reviewer for their valuable feedback. The provided suggestions are extremely helpful and constructive, and we will revise the paper accordingly. We address the reviewer's questions as follows.
>
> >Q1: The experiments focus on vision and NLP tasks. Have you tested ProDistill on other domains, such as speech or reinforcement learning? If not, do you believe the method would generalize well to these areas?
>
> **A1:**  Thank you for the great question. Most existing works on model merging primarily evaluate their methods on vision and NLP tasks, with limited exploration in other domains. Actually, we **can not even find a commonly used model merging benchmark for speech or reinforcement learning in our pretrain-finetune paradigm**.  Constructing such a benchmark would require significant effort in dataset selection, model training, and evaluation setup, which is unfortunately beyond the scope of the limited time available for the rebuttal.
>
> However,  we believe ProDistill can generalize well across different domains. The key reason is that **ProDistill is inherently domain-agnostic**—it does not leverage specific properties of vision or language data for its effectiveness. The only requirement is a layer-wise network structure, which is a common standard in modern deep learning.
>
> Besides, we would like to highlight that the LLMs used in our experiments, such as WizardMath-13B, are actually **trained using RLHF-based algorithms**. This can partly give evidence that ProDistill applies to RL trained models.
>
>
> >Q2: Will you release the code and implementation details for ProDistill?
>
> **A2:** Yes, the code and implementation details are already provided in the zip file of the Supplementary Material part on the openreview console.
>
> >Q3: While layer-wise distillation reduces memory overhead, are there any trade-offs in terms of performance or training stability compared to end-to-end distillation?
>
> **A3:** Thank you for the question. We compare ProDistill with its end-to-end distillation counterpart, **DistillMerge, in Appendix C.2**. In our experiments, we do not observe degradation in performance or training stability when using layer-wise distillation.
>
> Finally, we thank the reviewer once again for the effort in providing us with valuable and helpful suggestions. We will continue to provide clarifications if the reviewer has any further questions.

---

### Official Review · Reviewer_sgse · 2025-03-12

**Overall Recommendation:** 4

**Summary:**

The paper presents a new method for model merging based on progressive feature alignment. It proposes to learn merging coefficients by progressively aligning the representation of the merged model and the constituent models (the finetuned ones) layer by layer. This reduces the computational requirements of the end-to-end implementation.

**Claims And Evidence:**

All empirical claims are supported by convincing experiments.  See below for the theoretical part.

**Essential References Not Discussed:**

see above.

**Experimental Designs Or Analyses:**

The experimental analysis is well conducted. Missing comparisons in table 1,2 for data-agnostic methods is TIES merging [1], while for the data-driven approaches, it is important to include also MaTS [2]. In general, performances are impressive.

[1] Yadav, Prateek, Derek Tam, Leshem Choshen, Colin A. Raffel, and Mohit Bansal. "Ties-merging: Resolving interference when merging models." NeurIPS 2023
[2] Derek Tam, Mohit Bansal, Colin Raffel “Merging by Matching Models in Task Parameter Subspaces” TMLR 2024

**Methods And Evaluation Criteria:**

Method and evaluation setting are sound. The benchmark used are standard in the literature.

**Other Comments Or Suggestions:**

see above

**Other Strengths And Weaknesses:**

The main strength is that this is an actual scalable method.
Main weakness is that the theoretical motivation for this method seems irrelevant, considering the empirical performance of data-agnostic model merging methods, but I want to hear a better explanation from the authors.

**Questions For Authors:**

Could you give more details on the "dual inputs", I don't think I understood if this is something that is particular to your method or it just identifies the input of the merged and the fine-tuned models. The way is introduced is confusing as it seems a technical contribution, but to me this is just an implementation detail of your method.

**Relation To Broader Scientific Literature:**

I think the papers connect to the right literature on model merging, both data-driven and data-agnostic, building on top of insights from other papers that show that representation alignment is crucial for model merging.

**Theoretical Claims:**

The paper builds its motivation based on Theorem 3.1 and 3.2. These are the two worst-case analyses that the authors use in support of the motivation of their method, which instead is data-driven. Honesty, considering the success of data-agnostic methods and their empirical performance, I don’t think that these results are particularly motivating. This contribution seems irrelevant to the paper and the method itself.

---

> ### Author Rebuttal · Authors · 2025-03-31
>
> We would like to express our gratitude for the reviewer's helpful and positive comments. The suggestions provided have been instrumental in refining our work, and we will incorporate the necessary revisions accordingly. Below, we address each of the reviewer’s questions in detail.
>
> >Q1: Honesty, considering the success of data-agnostic methods and their empirical performance, I don’t think that these results are particularly motivating. ... Honestly, I think that these ablations and the results are much more insightful than the theoretical motivation. I suggest expanding them and finding space in the main paper.
>
> **A1:** Thank you for your thoughtful suggestion. We acknowledge that our theoretical results focus on worst-case analysis and may not fully capture real-world practice. However, we find these results delicate and interesting, as they highlight aspects that are not widely recognized in the community, which is why we included them in the main text.
>
> We agree the ablations in the appendix convey much information that we would like to share with the readers. We will restructure the paper to incorporate these results into the main text in future revisions.
>
> >Q2: Missing comparisons in table 1,2 for data-agnostic methods is TIES merging [1], while for the data-driven approaches, it is important to include also MaTS [2].
>
> **A2:** Thanks for the suggestions. We **add additional experiments to compare with TIES and MaTS**. For TIES, we use the same hyperparameter grid as in our LLM experiments. For MaTS, we use the RegMean Objective and RegMean Initialization, and choose the same hyperparameter grid as that of RegMean in our paper.
>
> The results are given below. Our method ProDistill outperforms the new baselines. An interesting finding is that TIES merging has a bad performance on NLP tasks, which coincides with previous findings in [1].
>
> |  | SUN397 |Cars |RESISC45| EuroSAT |SVHN |GTSRB |MNIST |DTD | Avg |
> |----------|----------|----------|----------|----------|----------|----------|----------|----------|----------|
> | TIES Merging    | 65.05     | 64.43     | 74.87     | 77.33     | 81.28     | 69.37     | 96.53    |54.52     |72.92    |
> | MaTS    | 67.65 | 67.23 | 81.76 | 93.30 | 86.76 | 80.08 | 97.39 | 60.21 | 79.30    |
> | ProDistill(Ours)    | 68.90 |71.21 |89.89 |99.37 |96.13 |95.29 |99.46 |68.03 |86.04|
>
>
> |  | CoLA| SST-2| MRPC |STS-B |QQP| MNLI| QNLI| RTE| Avg|
> |----------|----------|----------|----------|----------|----------|----------|----------|----------|----------|
> | TIES Merging    | 0.2050     | 0.8853    | 0.5427    | -0.2247     | 0.8282     | 0.7786     | 0.8142    | 0.6462     |0.5594    |
> | MaTS    | 0.3022 | 0.9243 | 0.8205| 0.5187 | 0.8178 | 0.7095 | 0.8506|  0.6498| 0.6992|
> | ProDistill(Ours)    | 0.4442| 0.9312 |0.8464 |0.6942| 0.8134| 0.7857| 0.8900| 0.7076|0.7641 |
>
>
> >Q3: I would have expected that DistillMerge would be an upper bound on the performance of ProDistill. Could you elaborate more on why it is not so? See Fig.5.
>
> **A3:** Thanks for pointing this out. We also find this result very intriguing. We hypothesize that ProDistill decomposes the overall training objective of DistillMerge, into more fine-grained layer-wise objectives. This decomposition makes it easier to minimize the objectives sequentially rather than optimizing the overall objective in one step.
>
>
> >Q4: Also in the right figure (Roberta exp) caption is DirectDistill. If this is not a typo, the caption should be updated.
>
> **A4:** Thanks for pointing this out. We will correct the typo in the revision.
>
> >Q5: Could you please give more details on DirectDstill? Am I understanding correctly that this baseline is just distillation where you fine-tune the model parameters of the merged model, instead of the lambda coefficients?
>
> **A5:** Yes exactly! The DirectDistill baseline uses $\ell_2$ distillation loss to fine-tune the original model parameters, instead of lambda coefficients.
>
>
> >Q6: Could you give more details on the "dual inputs", I don't think I understood if this is something that is particular to your method or it just identifies the input of the merged and the fine-tuned models. The way is introduced is confusing as it seems a technical contribution, but to me this is just an implementation detail of your method.
>
> **A6:** Thanks for the question. The proposed "dual inputs" is an important implementation detail in our method, which has a big impact in the final accuracy (see Appendix C.4). Additionally, this design choice is often overlooked in activation-matching algorithms, and we highlight it to ensure awareness within the community.
>
> Once again, we appreciate the reviewer's feedback and hope that our responses clarify the questions. We remain committed to improving the quality of our paper and welcome any further feedback.
>
> **References**
>
> [1] He Y, Hu Y, Lin Y, et al. Localize-and-stitch: Efficient model merging via sparse task arithmetic

---

### Official Review · Reviewer_CuV6 · 2025-03-13

**Overall Recommendation:** 3

**Summary:**

Model merging is an emerging paradigm that combines multiple models into **a single, versatile model, eliminating the need for extensive retraining and substantial weight storage**. However, it is commonly observed that the performance of merged models degrades as the number of models increases. To mitigate this, existing research has proposed using **a small amount of validation data** from downstream tasks to improve performance.

This paper starts with a theoretical analysis of data-free merging algorithms, showing that their performance is arbitrary and often suboptimal. And this analysis highlights that a small amount of few-shot data is **essential** to achieve meaningful, non-trivial performance. Inspired by these insights, the authors reframe model merging as **a distillation process**. They propose aligning the activations of each layer individually to boost the merged model's performance. This approach offers multiple efficiency benefits, including **reduced data, computation, and memory requirements**. Empirically, the authors validate their framework across three distinct experimental setups, successfully demonstrating its effectiveness from various perspectives.

**Claims And Evidence:**

1. The claims in this paper are clear and well-written, and the experimental results appear **convincing**. The proposed method offers **numerous efficiency benefits**—such as reduced data, computation, and memory usage—further demonstrating its practical applicability.

2. While the merging framework presented in this paper is promising, it consistently assumes that the models being merged share the same base architecture. What happens if the fine-tuned models were trained **under different configurations**, such as varying architectures, hyperparameters (e.g., batch size, learning rate), or training durations (e.g., total epochs)? **Misalignment prior to merging** could negatively affect performance, and this issue cannot be overlooked. Can the authors justify how your method handles alignment before merging models in such scenarios?

3. Furthermore, how does the proposed ProDistill method compare to **multi-task learning (MTL)** when both are trained on a validation dataset? What are the performance gap between these approaches?

**Essential References Not Discussed:**

There is no major reference to be discussed. However, it is worth noting that existing literature, such as the work on model alignment prior to merging (e.g., https://arxiv.org/abs/2209.04836), has explored layer-wise alignment techniques.

**Experimental Designs Or Analyses:**

Yes. Their experimental design is valid to demonstrate their method's effectiveness.

**Methods And Evaluation Criteria:**

The benchmark datasets and experimental setup used in the paper are commonly used in the model merging field. The evaluation makes sense to me and the analysis of efficiency from multiple perspectives looks solid to me.

**Other Comments Or Suggestions:**

In Equation 2, the method minimizes the feature distance between layer embeddings, which assumes that the embedding sizes of the models being merged are **identical**. How can this approach be extended to accommodate models with different embedding sizes, and what strategies could mitigate this limitation?

**Other Strengths And Weaknesses:**

This paper is easy to read, and their experimental results and framework efficiency look very promising.

**Questions For Authors:**

How do the coefficients for each layer and downstream task evolve across training steps? Are there insights from the results that could enable us to predict these coefficients **without performing the full training process**? Additionally, how do these coefficients relate to the correlation or similarity between downstream tasks? And more, is it necessary to compute coefficients for all the layers, can we reuse several close layers' coefficients without computing them?

**Relation To Broader Scientific Literature:**

The primary contribution of this work compared to other model merging techniques lies in **reframing the merging process as a distillation task**, leveraging a few-shot validation dataset, and progressively optimizing coefficients across model layers **in an efficient manner**.

**Theoretical Claims:**

I have checked their proof in Appendix A. Both of them look correct to me.

---

> ### Author Rebuttal · Authors · 2025-03-31
>
> We sincerely appreciate the reviewer’s valuable feedback. **We follow the reviewer's advice and conduct additional experiments, with the results provided at https://anonymous.4open.science/r/Experiments-for-Reviewer-CuV6-8476.** We address the reviewer's speficic questions as follows.
>
> >Q1: What happens if the fine-tuned models were trained under different configurations, such as varying architectures? ... How can this approach be extended to accommodate models with different embedding sizes?
>
> **A1:** Thank you for this insightful question. Our method can accommodate training differences such as varying hyperparameters; however, it is not directly applicable to merging models with different architectures, as activation matching strategies may fail. The broader challenge of merging models with differing architectures, which we highlight in the Related Works section, is a valuable research direction but is beyond the scope of this paper. Notably, most existing methods based on task vectors struggle with such settings, as weight averaging fails when architectures differ.
>
> >Q2: Can the authors justify how your method handles alignment before merging models in such scenarios?
>
> **A2:** Our method actually aligns the representation of each layer, which effectively addresses misalignment in the considered fine-tuning setup. In cases of severe misalignment, e.g. merging independently trained models, our method can be applied **in parallel to existing alignment algorithms**. Specifically, one can first apply an off-the-shelf alignment algorithm like Git-rebasin [1], and then apply our ProDistill to complete the merging process.
>
> >Q3: How does the proposed ProDistill method compare to multi-task learning (MTL) when both are trained on a validation dataset? What are the performance gap between these approaches?
>
> **A3:** We have already provided such an ablation study in **Appendix C.3**, where we study an algorithm termed as **DirectTrain**, which conducts supervised multi-task training on the few-shot validation datasets. Our method significantly outperforms DirectTrain algorithm, as indicated in Figure 6.
>
> >Q4: It is worth noting that existing literature, such as the work on model alignment prior to merging, has explored layer-wise alignment techniques.
>
> **A4:** Good point! The Git-rebasin paper [1] also explores a layer-wise activation matching strategy. The major difference is that
> 1. they use activation matching to learn a **permutation matrix** that aligns the neurons
> 2. we use activation matching to learn the **merging coefficients**,  which directly determine the final merged model.
>
> we have cited [1] and will make further discussions in the revision.
>
> >Q5: How do the coefficients for each layer and downstream task evolve across training steps?
>
> **A5:** Thank you for this insightful question. We conduct additional experiments to track the change of merging coefficients . The results are provided in **Figure 1~3 in the link.**
>
> We make several observations:
> 1. The mean of merging coefficients remains stable around its initialization 0.3.
> 2. The standard deviation of merging coefficicents keep increasing during training.
> 3. The cosine similarities between coefficients keep decreasing during training.
>
> These results indicate that ProDistill captures the **fine-grained (Obs 2) and task-specific (Obs3) variation within each module**, rather than merely adjust the overall scaling (Obs 1). Therefore, unlike previous approaches that use scalar coefficients, our element-wise coefficients carry richer information and are unlikely to be easily predictable from data or model alone.
>
> >Q6: How do these coefficients relate to the correlation or similarity between downstream tasks?
>
> **A6:** We provide additional experiments measuring the cosine similarity between merging coefficients across different tasks. The results, provided in **Figure 4 in the link**, reveal that the coefficients of different datasets have uniformly low cosine similarity of about 0.15. However, some correlation exists; for example, the coefficients for MNIST and SVHN (both digit classification tasks) have a slightly higher cosine similarity of 0.2.
>
> >Q7: Is it necessary to compute coefficients for all the layers, can we reuse several close layers' coefficients without computing them?
>
> **A7:** Thanks for the suggestions. We follow the advice and use vanilla task arithmetic for layers where task vectors have smallest $\ell_2$ norm. The results are given in **Table 1 in the link**. It shows that skipping closer layers degrades performance. However, even when skipping 6 out of the total 12 layers, the average accuracy is still above 80%.
>
> Once again, we sincerely thank the reviewer for their constructive feedback, and we are eager to engage in further discussions to clarify any concerns.
>
> **References**
>
> [1] Ainsworth S K, Hayase J, Srinivasa S. Git re-basin: Merging models modulo permutation symmetries[J].

---

> > ### Comment · Reviewer_CuV6 · 2025-04-02
> >
> > Thank you for your response. I have carefully reviewed your explanations.
> >
> > 1. Regarding Table 1 in your attachment, I find it particularly interesting to see how the trade-offs play out when combined with Figure 1. The approach of reusing coefficients for merging to reduce computation time is compelling. Identifying layers with similar coefficients and reusing them could be an effective optimization strategy.
> >
> > 2. It’s also intriguing to explore how this work handles different bases for model merging and the combination of models in various structures. I look forward to seeing further developments in your future work.
> >
> > Thank you for your time and response.

---

### Official Review · Reviewer_ZoNc · 2025-03-16

**Overall Recommendation:** 3

**Summary:**

The paper introduces ProDistill, a model merging algorithm leveraging progressive layer-wise distillation. A key contribution is the use of merging coefficients that are the same size as the model weights, enabling a fine-grained control of the merging process through element-wise operations. ProDistill efficiently computes these coefficients by separately optimizing each layer, reducing memory requirements compared to traditional end-to-end methods. The authors show that this progressive layer-wise approach significantly improves merging performance, achieving state-of-the-art results.

## update after rebuttal

While the authors clearly addressed my question, I still find that the method lacks a compelling justification—especially given that it requires more computation and data, yet does not consistently outperform Task Arithmetic (while I acknowledge that other baselines also show similar limitations). Even after considering the rebuttal, this key concern remains unresolved. Therefore, I do not think the paper is strong enough for an accept at the level of a 4, and I will maintain my original score.

(I’d like to note that I am not an expert in this domain, so I defer to the AC and other reviewers with more experience in this area when making the final decision.)

**Claims And Evidence:**

While the results of ProDistill are consistently strong on vision and small-scale NLP tasks, the performance gains are less clear and more nuanced in large language model (LLM) tasks, which may indeed be the most significant scenario. For instance, in Tables 3, 9, and 10, the simplest baseline—Task Arithmetic, which does not require a few-shot validation dataset or additional training—achieves better results than ProDistill on certain tasks. Why does ProDistill not show a clear performance advantage over Task Arithmetic in these cases? Is it due to insufficient few-shot data (≤64 samples)? If so, how many validation samples would be necessary to clearly outperform this baseline?

Additionally, regarding Figure 4 (left), which illustrates a clear positive relationship between validation set size and performance in vision tasks, can we expect similar trends to hold consistently for LLM tasks? Specifically, how do we explain or interpret the anomaly in Table 9, where 32-shot performance is higher than 64-shot performance?

**Essential References Not Discussed:**

None

**Experimental Designs Or Analyses:**

The experiments utilize well-defined and widely recognized benchmark tasks and datasets.

**Methods And Evaluation Criteria:**

The proposed methods and evaluation criteria generally make sense and are well-justified for the problem at hand, except for the evaluation of Large Language Model (LLM) tasks, as previously discussed in the "Claims and Evidence" section.

**Other Comments Or Suggestions:**

None

**Other Strengths And Weaknesses:**

None

**Questions For Authors:**

It would be helpful if the authors explained why a different set of baselines was chosen specifically for the LLM tasks compared to the other experiments.

**Relation To Broader Scientific Literature:**

This work can directly inform future research on model merging, particularly in developing scalable methods under conditions with moderate unlabeled data availability.

**Theoretical Claims:**

There are no notable issues or concerns to highlight.

---

> ### Author Rebuttal · Authors · 2025-03-31
>
> We greatly appreciate the reviewer's comments and valuable suggestions. **We conduct additional experiments to clarify the reviewer's question, with the results given in https://anonymous.4open.science/r/Experiments-for-Reviewer-ZoNc-9701.** We address the reviewer's questions in more detail as follows:
>
> >Q1: While the results of ProDistill are consistently strong on vision and small-scale NLP tasks, the performance gains are less clear and more nuanced in large language model (LLM) tasks...Task Arithmetic, which does not require a few-shot validation dataset or additional training—achieves better results than ProDistill on certain tasks. Why does ProDistill not show a clear performance advantage over Task Arithmetic in these cases?
>
> **A1:** This is a very good question. Task Arithmetic indeed has a good performance in our LLM experiments, which we summarize is due to the following reasons:
> 1. Task Arithmetic is more effective for **larger models**. In our vision experiments, its gap to fine-tuned models is over 20% for small ViT-B-32 models but narrows to about 10% for larger ViT-L-14 models. This trend aligns with findings from [1], which suggest that larger models exhibit stronger kernel behavior during fine-tuning and are closer to linear models (see Section D.1 in [1]).
> 2. Task Arithmetic works better for merging **a small number of models**. As shown in [2], increasing the number of models leads to greater task interference, reducing the effectiveness of methods that do not explicitly address interference. Our LLM experiments involve merging only two models, whereas our vision and NLP experiments merge up to eight models.
> 3. Task Arithmetic can have a **less balanced accuracy across tasks**, which gives a false sense of high performance. For example, when merging WizardMath-13B and Llama-2-13B-Code-Alpaca models (results given in Table 3 in our manuscript), task arithmetic shows slightly better performance than ProDistill on math related benchmarks (0.6467 vs 0.6279 on GSM8k), but significantly worse performance on code related benchmarks (0.0840 vs 0.2239 on MBPP). In other words, although Task Arithmetic may perform better on some tasks, **ProDistill provides more consistent improvements**, as reflected in the normalized average metric.
>
>
> >Q2: Additionally, regarding Figure 4 (left), which illustrates a clear positive relationship between validation set size and performance in vision tasks, can we expect similar trends to hold consistently for LLM tasks? Specifically, how do we explain or interpret the anomaly in Table 9, where 32-shot performance is higher than 64-shot performance
>
> **A2:** Thank you for this question. We conducted additional experiments and found that the inconsistency arises from **randomness in the sampling of validation data**. To confirm this and further investigate data scaling, we repeated the experiments from Table 9 using three different random seeds, and **extend the validation shot to 128**. The averaged performance of ProDistill is given in the table below. The updated results demonstrate a clear positive correlation between validation shot count and merging performance.
>
>
> | | AlpacaEval 2.0 | gsm8k| hendrycks math | human eval |mbpp |Avg | Normalized Avg|
> |-|-|-|-|-|-|-|-|
> | ProDistill 16 shots   | 0.1124     | 0.6370	    | 0.1302    | 0.2317    | 0.3120	| 0.2847   | 0.8798  |
> | ProDistill 32 shots   | 0.1134	    | 0.6387    | 0.1366    | 0.2317    | 0.3080	| 0.2857   | 0.8885   |
> | ProDistill 64 shots    | 0.1136     | 0.6414    | 0.1338    | 0.2500      | 0.3030	| 0.2884   | 0.8929   |
> | ProDistill 128 shots    | 0.1058     | 0.6624    | 0.1264   | 0.2866      | 0.3100	| 0.2982   | 0.9002   |
>
> To mitigate the effect of randomness, we **rerun all the LLM experiments for ProDistill** using three random seeds. The updated results can be found at https://anonymous.4open.science/r/Experiments-for-Reviewer-ZoNc-9701.
>
> >Q3: It would be helpful if the authors explained why a different set of baselines was chosen specifically for the LLM tasks compared to the other experiments.
>
> **A3:** We change the baselines for LLM tasks due to **scalability constraints**. Many of the baselines in vision and NLP tasks are computation or storage heavy, making them unsuitable for merging LLMs. For example, Adamerging requries storing each of the fine-tuned checkpoints in memory, and Localize-and-Stich requires solving an end-to-end optimization problems. Our method ProDistill is designed to be scalable and efficient, making it well-suited for merging LLMs without such constraints.
>
> We once again thank the reviewer for their valuable feedback and insightful questions. Please let us know if any further clarifications are needed.
>
> **Reference**
>
> [1] Ortiz-Jimenez G, Favero A, Frossard P. Task arithmetic in the tangent space: Improved editing of pre-trained models
>
> [2] Yadav P, Tam D, Choshen L, et al. Ties-merging: Resolving interference when merging models

---

> > ### Comment · Reviewer_ZoNc · 2025-04-03
> >
> > Thanks for the authors' rebuttal. I will keep the positive score.

---

### Decision · Program_Chairs · 2025-05-01

**Decision:**

Accept (poster)

**Comment:**

This paper explores a new technical perspective for model merging: leveraging a small amount of data from downstream tasks with layer-wise knowledge distillation to improve model merging performance. The paper initially got positive scores (3,3,4,3) by three knowledgeable reviewers and another reviewer, who mostly recognized the motivation, the idea and the basic performance of the proposed method. Meanwhile, the reviewers also raised some concerns about 1) less significant gains on LLMs compared to vision transformers; 2) unconvincing connections of some theoretical analysis (theorem 3.1 and 3.2) to the claimed motivation; 3) more experiments to study the generalization of the proposed method; 4) improving ablations in clarification and discussion.

The authors provided detailed responses to these concerns, which were most recognized by four reviewers. Finally, all reviewers maintained their positive scores. The AC read the paper, the reviews, the rebuttal and the reviewers' feedback, and agree with reviewers' assessment. Therefore, I recommend to accept this paper. The authors are encouraged to carefully consider the reviewers' comments/suggestions and their rebuttal in the final paper revision.